# Agent-State Construction with Auxiliary Inputs

**Ruo Yu Tao[1], Adam White [1,2], Marlos C. Machado[1,2]**
**[1] Department of Computing Science, University of Alberta**
**[2] Canada CIFAR AI Chair, Alberta Machine Intelligence Institute (Amii)**
`{rtao3,amw8,machado}@ualberta.ca`

**Reviewed on OpenReview:** `https://openreview.net/forum?id=RLYkyucU6k`

## Abstract

In many, if not every realistic sequential decision-making task, the decision-making agent is not able to model the full complexity of the world. The environment is often much larger and more complex than the agent, a setting also known as partial observability. In such settings, the agent must leverage more than just the current sensory inputs; it must construct an agent state that summarizes previous interactions with the world. Currently, a popular approach for tackling this problem is to learn the agent-state function via a recurrent network from the agent's sensory stream as input. Many impressive reinforcement learning applications have instead relied on environment-specific functions to aid the agent's inputs for history summarization. These augmentations are done in multiple ways, from simple approaches like concatenating observations to more complex ones such as uncertainty estimates. Although ubiquitous in the field, these additional inputs, which we term *auxiliary inputs*, are rarely emphasized, and it is not clear what their role or impact is. In this work we explore this idea further, and relate these auxiliary inputs to prior classic approaches to state construction. We present a series of examples illustrating the different ways of using auxiliary inputs for reinforcement learning. We show that these auxiliary inputs can be used to discriminate between observations that would otherwise be aliased, leading to more expressive features that smoothly interpolate between different states. Finally, we show that this approach is complementary to state-of-the-art methods such as recurrent neural networks and truncated back-propagation through time, and acts as a heuristic that facilitates longer temporal credit assignment, leading to better performance.

## 1 Introduction

In reinforcement learning, an agent must make decisions based only on the information it observes from the environment. The agent interacts with its environment in order to maximize a special numerical signal called the reward. This problem formulation is quite general and has been used in several high-profile success stories, such as agents capable of achieving impressive performance controlling fusion reactors (Degrave et al., 2022), beating Olympians in curling (Won et al., 2020), and when navigating balloons in the stratosphere (Bellemare et al., 2020). An important feature of these problems is that the environment the agent is in—the real world— is much bigger than the agent itself. In this setting, the current observation from the data stream the agent experiences does not contain all the relevant information for the agent to act on, making the environment *partially observable*.

In such settings, the agent must leverage more than just its current observations, it must construct an internal state that summarizes its previous interactions with the world, sometimes known as history summarization. We refer to this internal state as the *agent state*. One way to *learn* agent-state functions is to leverage recurrent neural network architectures (Hausknecht & Stone, 2015; Vinyals et al., 2019; Degrave et al., 2022) to summarize an agent's history. These recurrent functions calculate *latent* states (also called hidden states), and *learn* the function used to summarize history. Another approach that has been used in real-life use cases of reinforcement learning to help resolve partial observability is to model predictive information about the

agent's uncertainty over its effectiveness (Won et al., 2020) or observations (Bellemare et al., 2020); allowing the agent to reason about what information the agent does *not* know. Explicitly learning and leveraging predictions (Rafols et al., 2005b) is another approach that has been considered for history summarization. Two popular approaches to this are predictive state representations (Littman et al., 2001) and general value functions (Sutton et al., 2011), which represent agent state with future predictions. All these different approaches for constructing agent-state functions have different limitations and assumptions, but the same purpose: embedding necessary information from observations by expanding the feature space into a richer class of features with *auxiliary inputs* to ameliorate the issues of partial observability for better decision making.

Feature expansion has been considered in many different contexts, and has been widely used and investigated for neural networks over the years. Early incarnations of neural networks used random projections in the first layer of the neural network as "associator features" to map inputs to random binary features and expand the input space (Block, 1962). Expanding the input space to specifically tackle time-series data has been considered in the prediction context, where a convolution over the history of inputs (Mozer, 1996) has been proposed as an approach to incorporating simple, non-adaptive forms of memory for neural networks. In reinforcement learning, feature space engineering and expansion for agent-state construction was widely used before the advent of deep reinforcement learning. Techniques range from tile coding (Sutton & Barto, 2018) to radial and fourier basis functions (Sutton & Barto, 2018; Konidaris et al., 2011), having even been applied to Atari games (Liang et al., 2016). The largest drawback of these feature expansion techniques is the fact that they are static—they do not adapt to the problem setting at hand.

In this work we consider how we might resolve different forms of partial observability by augmenting the inputs to a function approximator (i.e. a deep neural network) with feature expansion techniques for reinforcement learning. We revisit earlier formulations of explicit history summarization (Mozer, 1996), and connect these ideas with modern approaches to agent-state functions. We look to combine the simplicity and performance of simple feature expansion techniques with the natural adaptivity and flexibility of neural network function approximation.

In this paper, we explore the idea that many approaches to tackle partially observable problems can be viewed as a form of auxiliary input. In the context of agent-state construction for reinforcement learning in partially observable environments, we define auxiliary inputs as *additional inputs, beyond environment observations, that incorporate or model information regarding the past, present and/or future of a reinforcement learning agent.* Auxiliary inputs have been ubiquitous across recent, real-world applications of reinforcement learning. Recent work in stratospheric superpressure balloon navigation with deep reinforcement learning has explored using not only the average magnitude and direction of the observed (or predicted) wind columns over time as input features, but also the variance of this wind column as an auxiliary input for successfully navigating balloons (Bellemare et al., 2020). In robotic curling, distance errors from previous throws were used as features to help mitigate the partial observability induced by environment conditions such as changing ice sheets over time (Won et al., 2020). In biomedical applications, both a time-decayed trace of joint activity (Pilarski et al., 2012) and future predictions of prosthesis signals (Pilarski et al., 2013) were used as additional input features for controlling or aiding in the control of robotic prostheses. All these approaches were successful due to carefully thought out auxiliary inputs that were fitting for their respective domains.

We present the following contributions in this work: (1) we survey *auxiliary input* techniques used for resolving partial observability in successful real-world reinforcement learning applications. (2) We unify these approaches under a single formalization for auxiliary inputs. (3) We demonstrate empirically, through illustrative examples, how a practitioner might leverage this formalism to create auxiliary inputs, and the efficacy of these approaches in de-aliasing states for better value function learning and policy representation.

We first introduce a formalism for the auxiliary inputs used throughout reinforcement learning. This formalism gives practictioners a mechanism to parse the aspects of a partially observable environment an agent may need to consider for both learning a value function and a successful policy, and as well as the *type* of auxiliary input to use in a given partially observable domain. We use a simple partially observable environment to elucidate how few simple, fast, and general instantiations of auxiliary inputs (a decaying trace of observations, particle filters, and likelihoods as predictions) summarizes history, as well as general trajectory information needed for decision making. Through demonstrations on this environment, we show that auxiliary inputs allow an agent

to discriminate between observations that would otherwise be aliased, and also allow for a smooth interpolation in the value function between different states. Next, we demonstrate the efficacy of uncertainty-based auxiliary inputs on two classic, partially observable environments. Finally, we show that particular auxiliary inputs (specifically exponential decaying traces) can integrate well with recurrent neural networks trained with truncated backpropagation through time (T-BPTT), potentially allowing for a significant performance increase as compared to using only one or the other. Code and implementation for this work is publically available[1].

To summarize, auxiliary inputs can be simple, performant, and should likely be the first approach most reinforcement learning practitioners take to tackle partial observability. In many cases, simple auxiliary inputs may be good enough if not better than more complex approaches such as recurrent function approximation, and they can also be easily combined with these approaches for better performance.

## 2 Background and Notation

We model the agent's interaction with the world as a sequential decision-making problem. On each time step $t$, the agent takes an action $a_t \in \mathcal{A}$ and receives an observation of the environment $o_t \in \mathcal{O}$. Partially in response to the agent's taken action, the environment transitions into a new state $s_{t+1} \in \mathcal{S}$ and receives a reward $r_{t+1} \in \mathcal{R}$, both according to the dynamics function $p : \mathcal{R} \times \mathcal{S} \times \mathcal{A} \times \mathcal{S} \to [0, 1]$. The agent does not observe the state, only the current observation, which we specify as a vector in this work[2] $o_t \in \mathcal{O} \subset \mathbb{R}^n$. This observation vector is constructed from the underlying state $S_t$ according to an unobservable function $o : \mathcal{S} \to \mathcal{O}$, where $o(S_t) \doteq O_t$. Periodically the environment enters a terminal state $S_L = \perp$ resetting the environment to a start state $S_0$. The agent's interaction is thus broken into a sequence of *episodes*.

The agent's primary goal is to learn a way of behaving that maximizes future reward. In our setting, the policy is defined over the history of interactions because the agent cannot observe the underlying state. Let $h_t \doteq \{O_0, A_0, O_1, ..., O_t\} \in \mathcal{T}$ where $\mathcal{T}$ denotes the space of all possible trajectories of observations and actions of all possible lengths. The return from timestep $t$ is the discounted sum of rewards $G_t \doteq R_{t+1} + \gamma R_{t+2} + ... + \gamma^{L-1} R_L$, where $\gamma \in [0, 1)$ is the discount and $L$ is the (stochastic) time of termination. The goal is to find a policy $\pi : \mathcal{T} \times \mathcal{A} \to [0, 1]$ that maximizes the expected return, in expectation across start states $\mathbb{E}_\pi[G_0]$. In this paper, we focus on methods that estimate a value function, $q_\pi(s, a) \doteq \mathbb{E}_\pi[G_t \mid S_t = s, A_t = a]$, in order to incrementally improve the agent's current policy $\pi$.

In our work we primarily focus on the Sarsa (Rummery & Niranjan, 1994) algorithm to learn estimates $q_\pi$ from the agent's interaction with the environment for control. Our policy $\pi$ is defined by two cases: either we choose a greedy action with respect to $q_\pi$ or we choose a random action with probability $\epsilon$ to ensure sufficient exploration. After sampling and taking an action $A_t \sim \pi(\cdot \mid S_t)$, we receive the next reward and next state $R_{t+1}, S_{t+1} \sim p(\cdot, \cdot \mid S_t, A_t)$, and pick the next action $A_{t+1} \sim \pi(\cdot \mid S_{t+1})$. In the more general, function approximation case, the estimate $\hat{q}_\pi$ is parameterized by $\boldsymbol{\theta}_t \in \mathbb{R}^k$ and is updated with the semi-gradient Sarsa update $\boldsymbol{\theta}_{t+1} \doteq \boldsymbol{\theta}_t + \alpha \left[ R_{t+1} + \gamma \hat{q}_\pi(S_{t+1}, A_{t+1}, \boldsymbol{\theta}_t) - \hat{q}_\pi(S_t, A_t, \boldsymbol{\theta}_t) \right] \nabla \hat{q}_\pi(S_t, A_t, \boldsymbol{\theta}_t)$, where $\alpha > 0$ is the step size, and $\nabla$ is the gradient of the function $\hat{q}$ with respect to the parameters $\boldsymbol{\theta}_t$.

In many problems, learning policies over full histories is not tractable and the agent must make use of an agent-state function to summarize the history. The *agent-state* function maps a given $h_t$ to an agent state vector $\boldsymbol{x}_t \in \mathcal{X}$. In most approaches to agent-state construction, including recurrent neural networks and general value function networks (Schlegel et al., 2021), the agent-state function has a recursive form:

$$\boldsymbol{x}_{t+1} \doteq u_\phi(\boldsymbol{x}_t, a_t, \boldsymbol{o}_{t+1}) \in \mathbb{R}^k, \tag{1}$$

where $\boldsymbol{\phi} \in \mathbb{R}^b$ are the parameters of the parametric agent-state function $u_\phi$. We can easily extend Sarsa to approximate the value function from agent state: $\hat{q}(\boldsymbol{x}_t, a_t, \boldsymbol{\theta}_t) \doteq \boldsymbol{\theta}^T \boldsymbol{x}_t \approx q_\pi(s_t, a_t)$. Thus, the estimated value is a linear function of the agent-state, which itself is a recurrent, potentially non-linear, function of $\boldsymbol{x}_t, a_t$, and $\boldsymbol{o}_{t+1}$. Semi-gradient Sarsa simply adapts $\boldsymbol{\theta}$ and $\boldsymbol{\phi}$ from the agent's interaction with the environment in order to improve reward maximization, as before. The aim of this work is to investigate how including auxiliary inputs in agent-state construction (as input to $u$) can improve value-based reinforcement learning agents.

---

[1]https://github.com/taodav/aux-inputs
[2]We denote random variables with capital letters, and vectors with bolded symbols.

## 3 Auxiliary Inputs

In our work, we investigate the use of auxiliary inputs as an input into the agent-state function, and its implications in the different types of partially observability. As opposed to only summarizing the past history of experiences, auxiliary inputs also allows agents to summarize the present and/or future for decision-making. To do this, auxiliary inputs must summarize entire *trajectories*, which also includes potential future interactions, rather than only the agent's history. We denote these trajectory of an agent at time $t$ as $\mathrm{T}_t \doteq \{\boldsymbol{O}_0, A_0, \boldsymbol{O}_1, ..., \boldsymbol{O}_t, A_t, ..., \boldsymbol{O}_L\} \in \mathcal{T}$, where $L$ denotes the terminal time step of the trajectory. At the time step $t$, an agent will have only a partially realized trajectory, $\mathrm{T}_t \doteq \{\boldsymbol{o}_0, a_0, ..., \boldsymbol{o}_t, a_t, \boldsymbol{O}_{t+1}, ..., \boldsymbol{O}_L\}$, where observations including and before $t$ are actualized variables (denoted with lower case letters), whereas all future observations from $t + 1$ to $L$ are still random variables.

Let $\boldsymbol{M} : \mathcal{T} \to \mathbb{R}^N$ denote an auxiliary input as a function of the trajectory of the agent, which maps the trajectory $\mathrm{T}_t$ to a fixed-length vector. As the input of this function is variable with respect to trajectory length, and the output is of fixed length, the auxiliary input function acts as a *summarizing function* which maps entire trajectories into some fixed-length vector that summarizes particular aspects of the trajectory. For an auxiliary input at time $t$, we denote this as $\boldsymbol{M}(\mathrm{T}_t) \doteq \boldsymbol{M}_t$. With these additional auxiliary inputs, we re-define our agent-state function to include these auxiliary inputs:

$$\boldsymbol{x}_{t+1} \doteq u_{\boldsymbol{\phi}}(\boldsymbol{x}_t, a_t, \boldsymbol{o}_{t+1}, \boldsymbol{M}_{t+1}) \in \mathbb{R}^k. \tag{2}$$

We visualize this auxiliary input function, as well as the overall agent-environment interface in Appendix A. Since trajectories T are likely to include future, unobserved random variables, it is sometimes helpful to define auxiliary inputs as functions that explicitly separate observed events from the past and unobserved random variables from the future:

$$\boldsymbol{M}_t \doteq \boldsymbol{M}(\{\boldsymbol{o}_0, a_0, ..., \boldsymbol{o}_t, a_t, \boldsymbol{O}_{t+1}, ..., \boldsymbol{O}_L\}) \tag{3}$$
$$= \boldsymbol{M}_h(\{\boldsymbol{o}_0, a_0, ..., \boldsymbol{o}_t, a_t\}) + \mathbb{E}_{\boldsymbol{M}_f}[\{\boldsymbol{O}_{t+1}, ..., \boldsymbol{O}_L\}]. \tag{4}$$

Where $\boldsymbol{M}_h$ and $\boldsymbol{M}_f$ are history summarizing and future summarizing functions respectively. $\mathbb{E}_{\boldsymbol{M}_f}$ denotes the expectation of future trajectories, summarized by the function $\boldsymbol{M}_f$. We explicitly consider expectations here due to the unobserved random variables (future observations, actions, and terminal time step) past the current time step, in order to consider concrete auxiliary inputs that do not depend on unobserved, future random variables. Equation (4) makes the distinction between the actualized, observed time steps, and the random variables of the trajectory clear: from 0 to $t$ our auxiliary input function maps over observed, actualized variables, and from $t + 1$ to $L$ we take the expectation with respect to the sampling distribution over all potential future observations and actions. We also present a further specified formulation of auxiliary inputs as a convolution operation in Appendix B for correctness and further clarity. Finally, we note that while the function $u_{\boldsymbol{\phi}}$ is described as a recurrent function in the general case, most of the control algorithms used in this work (besides the agents explicitly using an LSTM with auxiliary inputs in Section 5) do not leverage recurrency. This is because in most cases in practice, this is not done since auxiliary inputs are enough in terms of de-aliasing state and adding enough state information. Furthermore, in this study, we do this to isolate and understand the affects of these auxiliary inputs on state de-aliasing for value function learning by itself.

Many auxiliary inputs can be defined by the function $\boldsymbol{M}$. To clarify how this formulation might be used, we show how frame stacking (Mnih et al., 2015), widely used in algorithms that succeed in the Arcade Learning Environment (Bellemare et al., 2013; Machado et al., 2018), fits into this formalism.

**Frame Stacking.** As an auxiliary input, frame stacking only considers the past four observations the agent has seen. This means that the auxiliary input is only defined by the function $\boldsymbol{M}_t = \boldsymbol{M}_h$, where $\boldsymbol{M}_h$ is the function that concatenates the past 3 observations and the current observation together:

$$\boldsymbol{M}_h(\{\boldsymbol{o}_0, a_0, ..., \boldsymbol{o}_t, a_t\}) \doteq \boldsymbol{o}_{t-3} \otimes \boldsymbol{o}_{t-2} \otimes \boldsymbol{o}_{t-1} \otimes \boldsymbol{o}_t$$

where the $\otimes$ operation represents the concatenation operation. This auxiliary input function produces a fixed-length vector $\boldsymbol{M}_t \in \mathbb{R}^{4n}$, where $n$ is the size of the observations. This is exactly the frame stacking technique ubiquitous throughout Atari-2600 experiments.

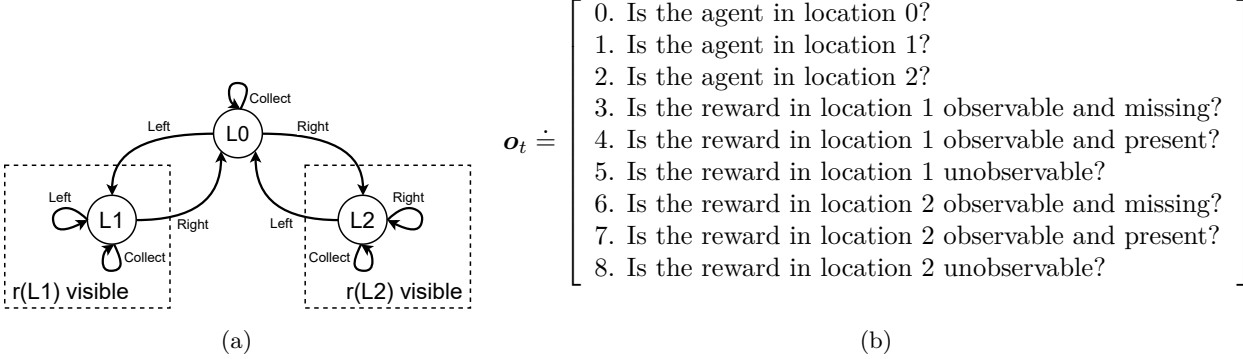

$$\boldsymbol{o}_t \doteq \begin{bmatrix} \text{0. Is the agent in location 0?} \\ \text{1. Is the agent in location 1?} \\ \text{2. Is the agent in location 2?} \\ \text{3. Is the reward in location 1 observable and missing?} \\ \text{4. Is the reward in location 1 observable and present?} \\ \text{5. Is the reward in location 1 unobservable?} \\ \text{6. Is the reward in location 2 observable and missing?} \\ \text{7. Is the reward in location 2 observable and present?} \\ \text{8. Is the reward in location 2 unobservable?} \end{bmatrix}$$

(a)          (b)

Figure 1: (a): The Lobster environment. (b) Binary questions representing the binary observation vector emitted by the environment at time $t$, $\boldsymbol{o}_t$.

**Resolution and Depth.** Viewing frame stacking as a form of auxiliary inputs elucidates an interesting property of our formalization: auxiliary inputs defined by Equation 3 represent differing *depths* and *resolutions* of the information you retain with regards to your trajectory. We define depth to be at what temporal length these auxiliary inputs retain information with regards to the agent's trajectory, and resolution to be the extent that information regarding individual observation-action pairs are preserved by the auxiliary inputs. Frame stacking is a form of auxiliary inputs that is low in depth (as we only see 3 time steps before the current time step), but high in resolution (since we retain all the relevant information of these 3 observations).

**Incremental Functions.** One important factor we focus on is the space and time complexity of calculating these auxiliary inputs. While we define these auxiliary inputs to be a function of history, we focus on algorithms that are incremental update functions for producing auxiliary inputs: $\boldsymbol{M}_t \doteq h(\boldsymbol{M}_{t-1}, \boldsymbol{O}_t, A_t)$. With this framework in place, we describe three auxiliary inputs in this section. Before we do so, we describe the minimal partially observable environment we use to illustrate the benefits of our auxiliary inputs on: the Lobster environment.

### 3.1 The Lobster Environment

We now introduce the Lobster environment: a simple partially observable environment we leverage to show the state de-aliasing properties of auxiliary inputs. The environment is shown in Figure 1a. In this environment, a fishing boat has to travel between 3 locations—represented as nodes in the graph—to collect lobsters from lobster pots. Only locations L1 and L2 have lobster pots, which refill randomly over time after being collected. The environment starts with both pots filled. The observations emitted by the environment include both the position of the agent, and whether or not a pot is filled if the agent is in the corresponding location. This observation vector is shown in Figure 1b. An agent in this environment has 3 actions: $\mathcal{A} \doteq \{\texttt{left}, \texttt{right}, \texttt{collect}\}$. The agent receives a reward of $+1$ if it takes the $\texttt{collect}$ action to collect the lobster pots at a location with full lobster pots. We fully specify the details of this environment in Appendix C.

### 3.2 Instantiations of Auxiliary Inputs

With this environment in place, we describe three techniques popular throughout reinforcement learning literature for auxiliary inputs with the formalism introduced in Section 3 that incorporate or model information from the past, present and/or the future. We consider these three techniques throughout our work. Through the demonstration of these auxiliary inputs in a small, partially observable domain called the Lobster environment, in this section we look to investigate and understand *how* these auxiliary inputs help with decision making in a simple and controlled setting. We look to compare the performance of each form of auxiliary input to two agents: one using only the observations described in Figure 1b, another using the fully observable environment state.

### 3.2.1 Exponential Decaying Traces

To incorporate information from the past of the agent, we consider exponential decaying traces of history as auxiliary inputs to agent-state. Decaying traces simply keep an exponentially decaying weighted sum of our observations and actions:

$$\boldsymbol{M}_t \doteq \sum_{\tau=0}^{t} \lambda^{t-\tau} g(\boldsymbol{o}_\tau, a_\tau) \tag{5}$$

with $\lambda < 1$, and $g$ is a preprocessing function for the observation, action pair. Written in an incremental form, we have $\boldsymbol{M}_t \doteq \lambda \boldsymbol{M}_{t-1} + g(\boldsymbol{o}_\tau, a_\tau)$.

This form of auxiliary input acts as a model of the past by acting as an exponential timer for events in the observation. When used as an auxiliary input to the agent-state function, an exponential decaying trace of observations allows the agent to take into consideration the time in which events occur in the observation vector $\boldsymbol{o}$ or action $a$. This particular form of history summarization is high in depth, but low in resolution, since $g(\boldsymbol{o}_\tau, a_\tau)$ is aggregated together across time steps.

### 3.2.2 Approximate Belief State with Particle Filters

We now consider a classic approach to auxiliary inputs for resolving partial observability: constructing approximate belief states. In this section, we investigate the use of uncertainty as auxiliary inputs—approximating a distribution over all states at every time step. In order to get a measure of uncertainty of the environment state, we leverage a few assumptions with regards to available transition dynamics and known state structure for the particle filtering approach for approximating a distribution over possible states, or a belief state (Kaelbling et al., 1998).

To construct these belief states, we consider a Monte-Carlo-based approach with particle filtering (Kitagawa, 1996) to approximate a distribution over states (Thrun, 1999; Pineau & Gordon, 2007) as auxiliary inputs for agent state. With this approach, we maintain an approximate distribution over possible states by incorporating statistics of the current observation and action into this distribution. We do this by approximating this distribution with particles and corresponding weights, and updating these particles with the emission probabilities and dynamics function through a particle filtering update. We begin with $k$ particles, which we denote by a vector of particle states $\hat{\boldsymbol{s}}_0 \in \{1, \ldots, |\mathcal{S}|\}^k$, initialized according to the start state distribution of the environment, and instantiate a vector of weights $\boldsymbol{w}_0 \in \mathbb{R}^k$. At every step $t+1$, and for every particle $\forall j \in \{1, \ldots, k\}$, with the dynamics function $p$, we update the particles and weights by first propagating all particles forward with the action taken:

$$\hat{\boldsymbol{s}}_{t+1}[j] \sim p(\cdot \mid \hat{\boldsymbol{s}}_t[j], A_t), \tag{6}$$

and updating each particle's weight according to the probability of emitting the observation (emission probability) received:

$$\overline{\boldsymbol{w}}_{t+1}[j] \doteq P\{\boldsymbol{O}_t = \boldsymbol{o}_t \mid S_t = \hat{\boldsymbol{s}}_t[j]\} \cdot \boldsymbol{w}_t[j]. \tag{7}$$

This produces the unnormalized weights of all particles. We get our new set of weights by simply normalizing: $\boldsymbol{w}_{t+1} \doteq \frac{\overline{\boldsymbol{w}}_{t+1}}{\sum_{i=1}^{k} \overline{\boldsymbol{w}}_{t+1}}$. These weights are essentially the mechanism in which we summarize our past trajectory.

With this mechanism to update particles and weights, we form our auxiliary inputs based on these weights. These auxiliary inputs calculate our approximate distribution of states at time step $t+1$, $\boldsymbol{M}_{t+1}$, by summing over weights for each particle for a given state:

$$\boldsymbol{M}_{t+1} \doteq \sum_{j=0}^{k} \boldsymbol{w}_{t+1}[j] \odot \mathbb{1}_{[\hat{s}_t[j]]}. \tag{8}$$

Where the bolded $\mathbb{1}_{[s]}$ corresponds to the one-hot encoding of length $|\mathcal{S}|$, with a 1 at state $s$. In this case, our auxiliary inputs $\boldsymbol{M}_{t+1}$ are our agent-state function, and we have $\boldsymbol{x}_{t+1} \doteq \boldsymbol{M}_{t+1}$.

At every step, the auxiliary input function $\boldsymbol{M}$ is defined by Equation 8. Our current action and observation are incorporated into our auxiliary input by both the propagation of particles forward given an action, and

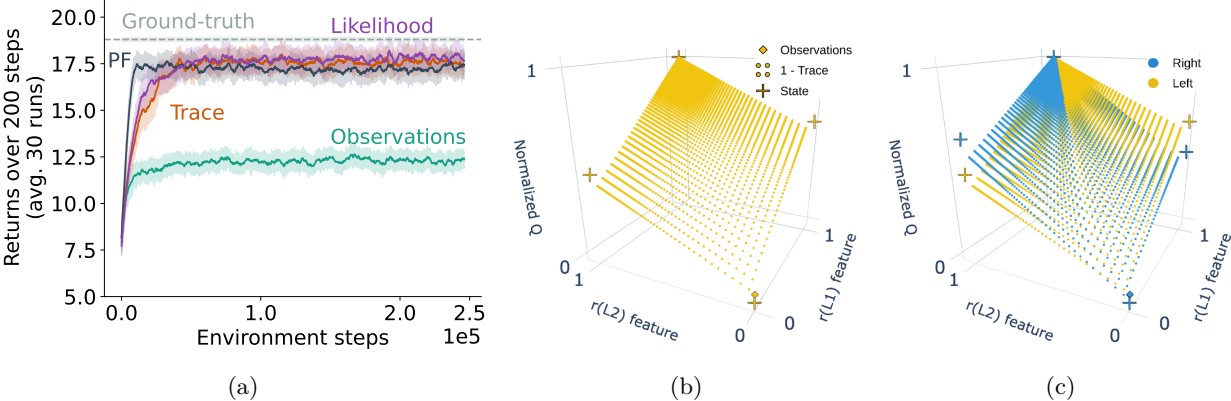

Figure 2: The impact of auxiliary inputs for agent states in the Lobster environment. Fig. 2 (a): Online returns in the Lobster environment for four different agents, in comparison to the optimal policy given the ground-truth state. Dotted line represents the optimal policy given states. See text for details. (b) Normalized action-values for an agent with decaying trace auxiliary inputs in location L0 for the `left` action. Each [r(L1), r(L2)] combination represents a possible input for the exponential decaying trace agent state. The values for the ground-truth state (crosses) and for using only the observation (diamond) are overlaid for comparison. See Appendix C for similar plots for particle filter and general value function auxiliary inputs. Fig. (c) Action-values for both `left` (yellow) and `right` (blue) actions in location L0.

the re-weighting of our particles through the emission probability of the current observation given the particle. The actions $a_t$ are incorporated into the particle updates for $\hat{s}_t[j]$ and observations $\boldsymbol{o}_t$ are incorporated into the weight updates for $\boldsymbol{w}_{t+1}$. Put together, $\boldsymbol{M}_t$ resolves partial observability through the counterfactual updates of the particles and their corresponding weights at the present time step with the current observation and action. This approach has low resolution because individual observations are incorporated into the approximate belief state through its emission probabilities. Its depth depends on the number of particles used.

### 3.2.3 Likelihoods for Incorporating Future Predictions and Past Information

We now consider how to incorporate future predictions together with past information for auxiliary inputs. In the reinforcement learning setting, one popular choice of predictions are general value functions (Sutton et al., 2011). These predictions take on the form of a discounted sum over *cumulants* (which can be a function over observations and actions), with some cumulant termination function. These can both be represented by our definition of auxiliary input functions $\boldsymbol{M}$.

We describe auxiliary inputs that incorporate both future predictions and past information for the Lobster environment. Let $i \in \{1, 2\}$ be the indices for both our two auxiliary inputs and our rewarding locations. These auxiliary inputs predict whether or not a reward at a given location L$i$ will be present if the agent takes the *expected* number of steps to that location. These auxiliary inputs essentially amount to answering the question: "Given that I saw r(L$i$) missing (r(L$i$) = 0) some steps ago, if I take the mean number of steps to reach L$i$, what is the likelihood that r(L$i$) will be regenerated (r(L$i$) = 1)?" We calculate these likelihoods by counting the number of time steps since seeing each reward, and the expected number of time steps to reach L$i$. We describe the full implementation of this auxiliary input in Appendix C.3.5.

Because this approach summarizes its history and future predictions with likelihood functions, it is low in resolution and high in depth. It is low in resolution because these likelihoods summarize its history into a probability distribution, and would be hard to recover individual observations. It is high in depth as exponential decaying traces are also high in depth: the likelihood probability distribution is also an exponential function of time steps, and so will have depth depending on the Poisson process rate.

### 3.3 Results on the Lobster Environment

We summarize the performance of the agents that leverage auxiliary inputs in Figure 2a. The policy learned with any of the three auxiliary inputs converges to a higher return than the agent using only observations. We can see this in the learned policies of our auxiliary-input agents versus the observations-only agent: when using only observations, the agent dithers between location L0 and location L1 *or* L2, but not both. On the other hand, the agent that leverages additional auxiliary inputs collects the reward from one of the rewarding locations, and then traverses to the other rewarding location depending on the value of the inputs.

Comparing the performance of all three auxiliary inputs in Figure 2a, we see the similarities in the converged average returns across the three algorithms. From all the visualizations of the action-value functions seen for the three auxiliary input approaches in Figures 2c, 7a, and 7b, we can see similarities in the learnt value functions between the three auxiliary inputs that incorporate or model different kinds of information. Learning a value function over these features result in similar policies. This implies that these auxiliary inputs all help resolve partial observability in one way or another. All hyperparameters swept and algorithmic details are fully described in Appendix C.2.

**Value Function Geometry of Auxiliary Inputs**  To get a better idea of *how* auxiliary inputs impact our policy, we consider the value function learnt over these auxiliary inputs. Specifically, since our value function is approximated with linear function approximation, we visualize and compare how the auxiliary inputs of exponential decaying traces affect the value function geometry of our learnt policy in Figs. 2b, and 2c. In these plots, we compare action-values when the agent is at location L0 between the three algorithms: the agent using observations only (the single diamond-shaped point), the agent using ground-truth environment states (the crosses), and finally the agent with exponential decaying traces as auxiliary inputs (the small circular points). We fully describe the details behind this 3D plot in Appendix C.4. We also show similar value function plots for both the particle filtering and likelihood auxiliary inputs in Figure 7 in Appendix C.3.

We first consider Fig. 2b; in this plot we calculate and visualize all possible input features mentioned above for the three algorithms over the normalized action-values of the `left` action, given the agent is at location L0. We can see that augmenting the agent state with auxiliary inputs *expands* the state space of the agent over two dimensions of time: one dimension for each reward observation that we have our exponential decaying trace over. The four vertices of this expanded state space represent and coincide with the four ground-truth states when the agent is at location L0.

Learning a value function with these exponential decaying traces allows the agent to smoothly interpolate between the values of the actual ground-truth states. The agent does this through resolving (checking out a location) or accumulating (waiting in another location) the decaying trace observations. Learning a value function with such agent states essentially allows the agent to disentangle and discriminate states that would otherwise be mapped to the same observation. This expansion in the state space allows for more expressivity in the value function.

This additional expressivity is also reflected in the policy. This is depicted in Figure 2c; in addition to the action-value function for the `left` action, we also visualize the action-value function for the `right` action of the agent with exponential decaying traces. By overlaying the action-values of both actions we can see how the decaying trace agent, as well as the agent using the ground-truth states, learn action-values that in some corners are greater for the `right` action and in other corners are greater for the `left` action, actually leading to a sensible greedy policy. Alternatively, the agent that uses only the environment observations has no choice but to collapse the action-values of both actions into the same value. Finally, we present results in Appendix C.5 on these same auxiliary inputs, but with a base control algorithm that can represent stochastic policies - Proximal Policy Optimization (Schulman et al., 2017). We show that while stochastic policies help in partially observable settings, auxiliary inputs resolve partial observability that is unresolvable by stochastic policies alone.

In this section, we have presented results and visualizations to help elucidate why these techniques help with decision making in partial observability. We have shown that auxiliary inputs expand the input space to de-alias states, and allow for more fine-grained policy representations and ultimately better performance. With these definitions in place, in this manuscript we further investigate two of these approaches for auxiliary inputs: present (particle filters) and past (exponential decaying traces) modelling.

# 4    Particle Filtering for Auxiliary Inputs

We first consider the case with stronger assumptions: particle filtering for auxiliary inputs. We know that this form of auxiliary input adds the maximal amount of relevant information (Kaelbling et al., 1998) to our agent state in the partially observable setting. With this approach, we look to answer the following in this section: how much can auxiliary inputs help? How does it compare to using recurrent neural networks for function approximation?

In this section, we look at auxiliary inputs which explicitly represent *uncertainty*. Uncertainty as auxiliary inputs has been used in many real-world reinforcement learning use cases, including wind-column uncertainty for stratospheric superpressure balloon navigation (Bellemare et al., 2020), and positional entropy as features for robotic navigation with reinforcement learning (Roy & Thrun, 1999). In this section, we consider particle-filtering-based uncertainty features for auxiliary inputs as described in Section 3.2.2. We evaluate this approach on classic partially observable environments: A modified version of the *Compass World* (Rafols et al., 2005a) environment and the *RockSample* (Smith & Simmons, 2004) environment. We compare this approach to agents using LSTMs for agent-state construction (Bakker, 2001). We begin by describing these two environments.

## 4.1    Compass World and RockSample

Modified Compass World is a $9 \times 9$ partially observable environment where the agent can only see the color of the square directly in front of it. The goal of the agent is to face the green square. The agent is initialized in a random position that is not the goal position. The agent has 3 actions: {`move forward`, `turn left`, `turn right`}. Due to this form of partial observability, the goal location, and the random start position and pose, the agent has to resolve both its $x$ and $y$ coordinates in order to reach the goal and receive a reward of $+1$. This environment is modified from the original Compass World environment in that the goal location is in the *middle* of the west blue wall, as opposed to the top of the west blue wall. This is to add difficulty to this environment; in the original environment, the agent would only need to traverse up to the north facing orange wall (resolve its y-position) and head west, until it reached the goal. In this modified version, the agent needs to resolve both coordinates for it to reach the goal. A visualization of this environment is shown in Figure 10a in Appendix D.

RockSample(7, 8) is another partially observable environment where the goal of the agent is to collect as many good rocks as possible before exiting to the eastern border of the gridworld. In this environment, we have a $7 \times 7$ gridworld with 8 rocks randomly scattered throughout the environment. At the start of an episode, each rock is randomly assigned to be either good or bad. The agent is unaware of whether or not a rock is good or bad, but has individual actions to check the goodness of each rock with an imperfect sensor that gets noisier the farther away the agent is to each rock. Besides these `check` actions, the agent also has the move actions {`up`, `right`, `down`, `left`}. Collecting a good rock gives a positive reward of $+10$, and exiting to the right also gives a positive reward of $+10$. Collecting a bad rock gives a reward of $-10$. A visualization of the Rock-Sample environment is shown in Figure 10b in Appendix D, along with further details of both environments.

## 4.2    Results and Discussion

We compare our particle-filter-based auxiliary inputs to other agent-state functions for Modified Compass World and RockSample in Figures 3a and 3b respectively. The particle filtering-based agent (orange, labelled as "Auxiliary Inputs") is compared to three baseline agents: An agent using only observations (teal, labelled as "Observations"), an agent which uses an LSTM to learn its agent-state function (yellow, labelled as "LSTM"), and an agent that uses the ground-truth environment state (blue, labelled as "Ground-truth").

In these figures, we plot the online returns during training over environment steps. All experimental results shown report the mean (solid lines) and standard error to the mean (shaded region) over 30 runs. Standard error to the mean is shaded but too small to be visible. Hyperparameters for each algorithm were selected based on a sweep. In these experiments, all agents utilize similar neural network architectures as their function approximator. Further details of the experimental setup and hyperparameters swept can be found in Appendix D. Further details of the particle filter and the agent-state function for each environment in this

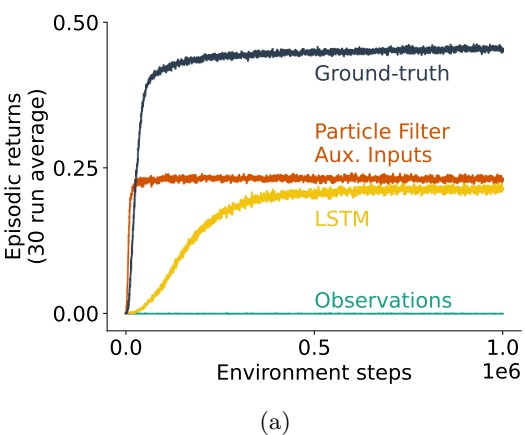 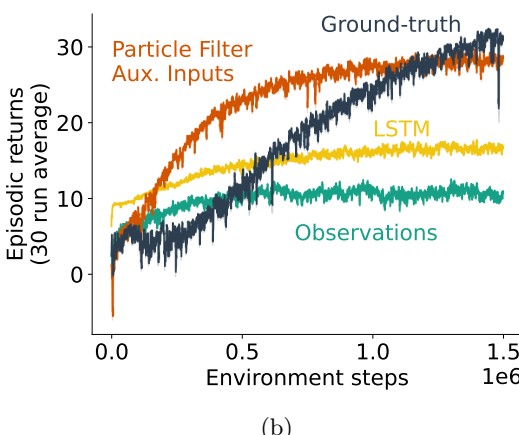

(a)                  (b)

Figure 3: Online discounted returns over environment steps for agents in both the Modified Compass World (3a) and RockSample(7, 8) (3b) environments. Colors in both plots correspond to the same class of agents.

section can be found in Appendix D.2. We also perform ablation studies over the RockSample half efficiency distance and LSTM-input action concatenation in Appendix D.3.

The agents acting on observations (and not ground-truth state) must first resolve their partial observability in both environments. We first consider the learnt policies from both the LSTM and particle filter agent states in Modified Compass World. The agents first move forward until they see a wall color. Seeing this wall color resolves one of their position coordinates, and allows the agent to navigate towards the west wall. The agent then traverses either up or down, periodically checking the color of the west wall until they can resolve the other position coordinate, and get to the goal. This resolving of coordinates is *implicitly* represented as elements of the distribution over state going to 0. In RockSample, using an approximate belief state as auxiliary inputs is particularly useful because these auxiliary inputs allow the agent to learn a policy that takes into account the uncertainty of the rocks in the current time step. The agent will traverse closer to a rock before checking whether the rock is good or bad since accuracy of the check decreases with distance.

The particle filter auxiliary input approach (with the additional assumptions that approximate belief states afford) also consistently outperforms the agent using an LSTM agent-state function. Not only do these auxiliary inputs converge faster, but also converges to a higher average return. This implies that these auxiliary inputs are able to represent and add privileged information into the agent state that may be hard to learn and represent in recurrent neural network approaches, and are also beneficial for faster value function learning.

Incorporating trajectory information with particle filtering allowed the agent to leverage knowledge of the dynamics of the environment. This results in auxiliary inputs that were highly relevant for decision making. While the particle filtering approximate belief state assumptions may be strong assumptions to make, it is the case in many real-world use cases that this information (or approximations thereof) are not completely unreasonable (e.g., Bellemare et al., 2020). In these experiments, we have shown that in the ideal case, auxiliary inputs are able to add complex, relevant information to the agent state given a model of the environment. We now consider the case where we do not have these assumptions available and investigate modelling the past for auxiliary inputs.

## 5 Scaling Up Auxiliary Inputs and Integration with RNNs

With the efficacy of auxiliary inputs demonstrated in smaller environments, in this section we now consider how to scale up approaches to auxiliary inputs, as well as the role of auxiliary inputs in gradient-based agent-state functions. Specifically, in this section we investigate ways in which to scale up exponential decaying traces described in Section 3.2.1 to larger, pixel-based environments, and how these traces integrate with recurrent neural networks as an agent-state function. We begin by describing the two variations of the partially observable environments we test on.

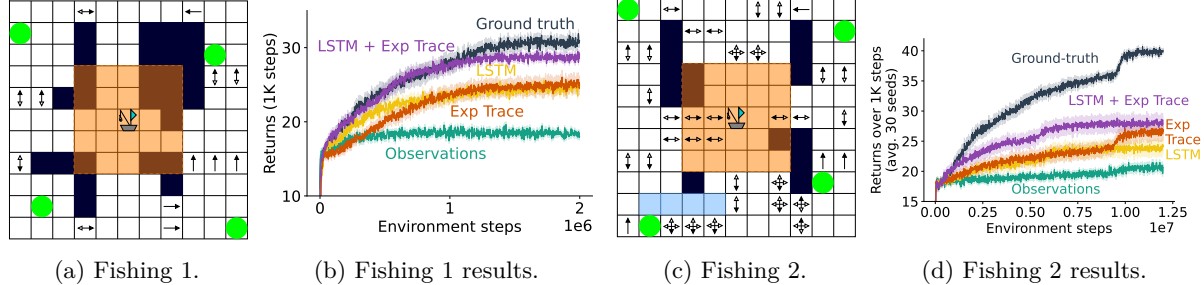

(a) Fishing 1.    (b) Fishing 1 results.    (c) Fishing 2.    (d) Fishing 2 results.

Figure 4: (a, c) The Fishing Boat environments. At every time step the agent receives a map with the $5 \times 5$ area around itself updated. This map includes obstacles, currents and rewards. Currents labelled with multiple directions represent stochastic currents. Dark blue grids represent obstacles, whereas light blue grids represent obstacles the agent is able to see through. (b, d) Results for the first and second Fishing Boat environments respectively, averaged over 30 runs. Standard errors are shaded for each curve.

## 5.1   The Fishing Environments

We first introduce the *Fishing* environments — two pixel-based stochastic and partially observable environments, visualized in Figures 4a and 4c. This environment is reminiscent of a scaled-up version of the Lobster environment, in that it was designed to test an agent's ability to reason about partial observability and time, except in a scaled-up setting. In both of these foraging environments, the goal of the agent (a fishing boat) is to continually navigate around the stochastic currents and obstacles to collect fish from fishing locations (denoted by the green circles). Currents push the agent one step in the direction it is facing. After collecting the fish from a net, the net is re-cast and fills up over a random amount of time. Partial observability in the domain comes from a few sources: the first source of partial observability is the environment map. At every step, the position of the agent is given and a map of the environment is accumulated into a larger map, much like in robot navigation and mapping (e.g., Elfes, 1987; Thrun, 1998). In a given step, the map is only updated with a potentially occluded $5 \times 5$ area around the agent's current position. This $5 \times 5$ area contains information on the direction of the currents, obstacles and rewards at the current time step. As the agent traverses the environment, current and reward information on the accumulated map unobserved by the agent begins to "stale" since currents and rewards change stochastically with time. Further details of this environment are elucidated in Appendix E.1.

Due to the stochasticity of the currents and rewards, the degree of partial observability of each environment is dictated by the number of stochastic elements throughout the map, and the rate at which these random variables change. In Fishing 1 (Figure 4a), we have an environment with low levels of partial observability and stochasticity. There is a sparse number of currents throughout the map, with most currents acting as a gateway to the rewarding areas. Rewards and currents in this environment also have a relatively slow rate of change. Fishing 2 (Figure 4c) is an environment that is much more partially observable, with several fast-changing currents throughout the environment. Specifics of the stochasticity in each environment are detailed in Appendix E.1.2.

## 5.2   Exponential Decaying Traces For Robot Mapping

To encode information with regards to the past history of the agent as an auxiliary input for these environments, we use exponential decaying traces (as per Section 3.2.1) over the past observable regions in the environment map. In this case we have auxiliary inputs that are all updated at once as a *matrix*, which we define as $\boldsymbol{M}_t \in \mathbb{R}^{d \times d}$, where $d$ is the width and height of the map. Our auxiliary input is then:

$$\boldsymbol{M}_t \doteq \sum_{\tau=0}^{t} \lambda^{t-\tau} \mathbb{1}(\boldsymbol{o}_\tau). \tag{9}$$

$\mathbb{1}(\boldsymbol{o}_\tau) \in \{0,1\}^{d \times d}$ is a $d \times d$ binary map which indicates which areas of the global map are observable from $\boldsymbol{o}_\tau$ at time $\tau$. At each step, all the locations that are not currently observable are decayed by a factor of $\lambda < 1$. This auxiliary input encodes the time since the agent has observed a particular location as an exponentially

decaying timer. The incremental version of this auxiliary input is simply $\boldsymbol{M}_t \doteq \max(\lambda \boldsymbol{M}_{t-1} + \mathbb{1}(\boldsymbol{o}_t), \mathbf{1})$, where $\mathbf{1}$ is a $d \times d$ matrix of ones.

**Results and Discussion** We show our results for both environments in Figures 4b and 4d. Experimental details including hyperparameters swept, algorithmic details, and environment details are included in Appendix E. Results shown here are offline evaluation returns over environment steps, where we evaluate our agent after every fixed number (10K) of steps. We compare our exponential decaying trace auxiliary inputs (orange) to a few baselines: an LSTM-based agent with action concatenation (yellow), and an agent with only the observation map as described before as input (teal), and a combination of both the trace auxiliary inputs combined with an LSTM agent-state function (purple). In both environments, exponential decaying traces as auxiliary inputs are comparable to, or performs slightly better than the LSTM-based agent. In the simpler Fishing 1, using exponential decaying traces matches the performance of the LSTM agent, as shown in Figure 4b. While the LSTM agent performs temporal credit assignment by using additional compute with T-BPTT and TD error propagation, our exponential decaying trace can be seen as a simple way of performing temporal credit assignment by only propagating TD error *through the input features*, reminiscent of eligibility traces (Sutton & Barto, 2018). Doing so requires much less computation per time step as compared to T-BPTT. In the Fishing 2 environment, the additional stochasticity seems to harm the performance of the LSTM agent as compared to the agent using exponential decaying traces as auxiliary inputs, with the decaying trace agent slightly outperforming the LSTM agent in this case.

Exponential decaying traces are also able to integrate *with* LSTMs. One might expect no increase in performance when combining the two approaches, since LSTMs are able to exactly model an exponential decaying trace of observation features. But from the results, we see that in both environments, combining decaying trace auxiliary inputs with LSTM function approximation increased the performance of the agent by quite a large margin in these environments. This implies that when combined, the trace features and LSTM hidden states are modelling different, but complementary, aspects of the partially observable environment for even better performance. As pointed out by Rafiee et al. (2022), adding an exponential decaying trace as input to an LSTM learning through T-BPTT seems to add robustness to the truncation window length. In our case, for control, it seems to both increase the rate of learning and to also increase the average returns of the learnt policy. This example suggests that auxiliary inputs can integrate well with gradient-based agent-state construction.

To conclude, in this section we have demonstrated the the ability of a decaying trace as an auxiliary input to scale to a larger environment, and the ability of this auxiliary input to integrate well with RNN agent-state functions.

## 6 Conclusion and Discussion

To conclude, auxiliary inputs are helpful tools for reinforcement learning practitioners to resolve partial observability which also have the potential to integrate well with existing gradient-based agent-state functions.

In this work we advocate for the general principle of auxiliary inputs as an addition to agent-state construction, and evaluate different instantiations of auxiliary inputs for reinforcement learning. We first introduce auxiliary inputs as a unifying framework for input augmentation, as well as consider three different instantiations of these inputs in Section 3. Using the Lobster environment introduced in Section 3.1, we demonstrate the efficacy of these auxiliary inputs in resolving partial observability, as well as how these auxiliary inputs allow us to *expand* the input feature space of the agent to allow us to interpolate between ground-truth states, and for a more fine-grained policy. With this formalism in place, we investigate the performance of the particle filtering approach to auxiliary inputs on a few classic partially observable environments in Section 4. We show the efficacy of approximate belief states as auxiliary inputs in these hard, partially observable environments. Finally, in Section 5, we investigate the use of simple exponential decaying traces of observation features as auxiliary inputs on the scaled-up pixel-based Fishing environment. Besides showing matching or better performance of these trace features in this environment as compared to LSTMs, we also show how this auxiliary input can *integrate* with recurrent neural network agent-state functions, and improve performance.

As for future work, the most immediate extension of our investigation would be an empirical study on the relative efficacies of different auxiliary inputs on different forms of partial observability and different learning algorithms on potentially larger environments. This extension could also consider a more fined-grained comparison of auxiliary inputs to learnt recurrent representations (e.g. LSTMs vs exponential decaying traces). Along this vein, another, tangential direction of future work would be to utilize the formalism of auxiliary inputs developed here as a guide for the (potentially automatic) selection of auxiliary inputs for a particular domain. Another avenue for future work would be to use more complex predictions as auxiliary inputs. In terms of using future predictions to resolve partial observability, general value functions (Sutton et al., 2011) and predictive state representations (Littman et al., 2001) are two promising approaches for using predictions for resolving partial observability as an auxiliary input.

While a promising area of research, future predictions have their limitations: in the context of using future predictions for next-step predictions as well as for control, general value functions have been shown to be effective in only a very limited scope of environments and predictions (Schlegel et al., 2021). Another very promising direction for future work would be to integrate and combine different forms of auxiliary inputs, which leverage information from all parts of the trajectory, from both past and future. Another promising direction for future work is to leverage auxiliary inputs for better exploration. While techniques to estimate the uncertainty of an agent for exploration are near ubiquitous throughout reinforcement learning, an interesting avenue for exploration is learning policies over altered *inputs* of an agent for more robust exploration.

### Acknowledgments

We would like to thank Martha White for the support with computational resources, ideas, and feedback. We would also like to thank Matthew Schlegel for the continued support throughout this project, especially in their help in developing predictive algorithms for the Lobster environment. We would like to thank Patrick Pilarski for the many insightful comments and suggestions that have improved this work. Finally, we would like to thank Marc G. Bellemare and Taylor W. Killian for early discussions on what is now auxiliary inputs.

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

# A    Auxiliary Input Function in the Agent Environment Interface

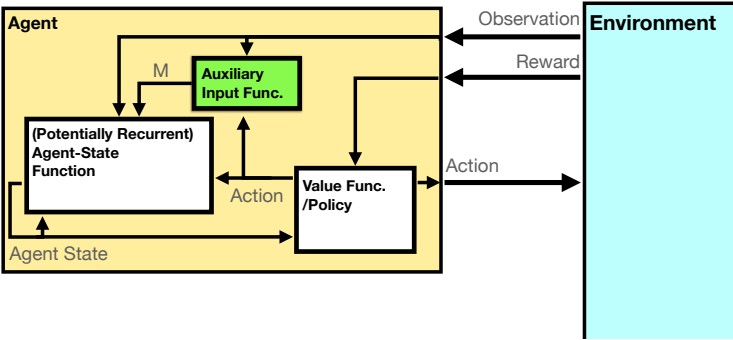

Figure 5: The agent-environment interface, with the auxiliary input function visualized within the agent.

# B    Auxiliary Inputs as Convolutions Over Trajectories

Another useful formulation and further specification of auxiliary inputs is to view them as a convolution operation over trajectories. While previous work has viewed *history* summarization as a convolution operation (Mozer, 1996), in this work we formulate the convolution over *trajectories* as auxiliary inputs which summarize both histories and potential futures.

In the more specified form, we can have multiple functions $\boldsymbol{m}^i, i \in \{0, \ldots, N-1\}$ which correspond to each of our auxiliary inputs over our history. For the $i$th auxiliary input at time $t$, we denote this as $\boldsymbol{m}^i(h_t) \doteq \boldsymbol{m}^i_t$. The set of auxiliary inputs at time $t$ is then written as the tuple $\boldsymbol{M}_t \doteq (\boldsymbol{m}^0_t, \ldots, \boldsymbol{m}^{N-1}_t)$.

We further specify the function $\boldsymbol{m}$ to allow us to formalize our different approaches to auxiliary inputs. Without loss of generality, we modify our definition of history to be a sequence of $(\boldsymbol{o}, a) \in \mathcal{O} \times \mathcal{A}$ observation-action tuples (we can simply append the next action $A_t$ or the empty set $\emptyset$ to the final observation): $h_t \doteq \{(\boldsymbol{O}_0, A_0), (\boldsymbol{O}_1, A_1), ..., (\boldsymbol{O}_t, A_t)\} \in \mathcal{T}$. With this, our $i$th auxiliary input at time $t$, $\boldsymbol{m}^i_t$, can be seen as a convolution over the history of preprocessed observation-action pairs:

$$\boldsymbol{m}^i_t \doteq \sum_{\tau=0}^{t} k_i(\tau) g_i(\boldsymbol{O}_\tau, A_\tau) \tag{10}$$

where $k_i$ is the $i$th kernel function $c : \mathcal{N} \to \mathbb{R}$ of the convolution, and $g$ is the preprocessing function applied to the observation-action pair before convolving with the rest of the history. Many auxiliary inputs and memory mechanisms can be defined by the function $\boldsymbol{m}$.

## B.1    Frame Stacking as an Auxiliary Input with Convolutions

As an auxiliary input, the number of previous frames to stack corresponds to the number of auxiliary inputs ($N = 3$, as the current frame is accounted for). Our preprocessing function $g$ for frame stacking is simply the function that just returns the observation: $g(\boldsymbol{o}_t, a_t) \doteq \boldsymbol{o}_t$. The kernel function for $i \in \{0, \ldots, 2\}$ is defined as

$$k_i(\tau) \doteq \mathbb{1}_{[\tau = t - i - 1]},$$

where $\mathbb{1}_{[cond]}$ is the indicator function, which is 1 if *cond* is true, 0 otherwise. With these definitions in place, Equation (10) defines 3 auxiliary inputs at time $t$, $\boldsymbol{M}_t \doteq (\boldsymbol{m}^0_t, \ldots, \boldsymbol{m}^2_t)$. Furthermore, based on the convolution of this time-indicator function, each of these inputs correspond to the observation $O_{t-i-1}$ — or the previous 3 observations seen by the agent. $\boldsymbol{M}_t$ defines the stack of the last 3 observations, which, in addition to the current observation, is frame stacking as described by Mnih et al. (2015).

### B.2 Exponential Decaying Traces with Convolutions

Decaying traces written as a convolution over a trajectory simply keep an exponentially decaying weighted sum of our trajectory observations and actions:

$$\boldsymbol{m}_t^i \doteq \sum_{\tau=0}^{t} \lambda^{t-\tau} g_i(\boldsymbol{o}_\tau, a_\tau), \tag{11}$$

with $\lambda < 1$. The kernel function $k_i(\tau)$ in this case is a simple exponential function over past time steps:

$$k_i(\tau) \doteq \begin{cases} \lambda^{t-\tau} & \text{if } \tau \leq t, \\ 0 & \text{otherwise.} \end{cases} \tag{12}$$

Our decaying trace auxiliary inputs are simply convolutions over time with this kernel function. Written in an incremental form, we have $\boldsymbol{m}_t^i \doteq \lambda \boldsymbol{m}_{t-1}^i + g_i(\boldsymbol{o}_\tau, a_\tau)$.

### B.3 Particle Filtering with Convolutions

Given the mechanism to update particles and weights in Section 3.2.2, we form our convolution-based auxiliary inputs based on these weights. In this approach, we only have a single auxiliary input vector where $N = 1$, which we simply denote as $\boldsymbol{m}_t$.

In this case, our convolution-based auxiliary inputs are the same as our simplified auxiliary inputs, with $\boldsymbol{m}_t = \boldsymbol{M}_t$, with the kernel function defined as:

$$k(\tau) \doteq \begin{cases} 1 & \text{if } \tau = t, \\ 0 & \text{otherwise.} \end{cases} \tag{13}$$

The preprocessing function $g$ is defined by $g(\boldsymbol{o}_t, a_t) \doteq \sum_{j=0}^{k} \boldsymbol{w}_{t+1}[j] \odot \mathbb{1}_{[\hat{s}_t[j]]}$.

### B.4 Likelihoods as Convolutions

To represent likelihoods and general value functions as convolutions, we simply represent our cumulant as a function of observations and actions like our preprocessing function $g_i$, with some separate cumulant termination function which is represented with our kernel function $k_i$.

## C Lobster Fishing Environment and Experiment Details

### C.1 Environment Details

As this environment is partially observable, we now detail the observation vector $\boldsymbol{o}_t \in \{0,1\}^9$ the agent receives at every time step. We list out 9 ordered true or false questions which correspond to the elements (either 0 or 1 respectively) in the observation vector:

$$\boldsymbol{o}_t \doteq \begin{bmatrix} \text{0. Is the agent in location 0?} \\ \text{1. Is the agent in location 1?} \\ \text{2. Is the agent in location 2?} \\ \text{3. Is the reward in location 1 observable and missing?} \\ \text{4. Is the reward in location 1 observable and present?} \\ \text{5. Is the reward in location 1 unobservable?} \\ \text{6. Is the reward in location 2 observable and missing?} \\ \text{7. Is the reward in location 2 observable and present?} \\ \text{8. Is the reward in location 2 unobservable?} \end{bmatrix} \tag{14}$$

We now detail the sources of stochasticity in the Lobster environment. Actions that try to transition between locations in the Lobster environment succeed with probability $p_{slip} = 0.6$; if the transition fails, the agent

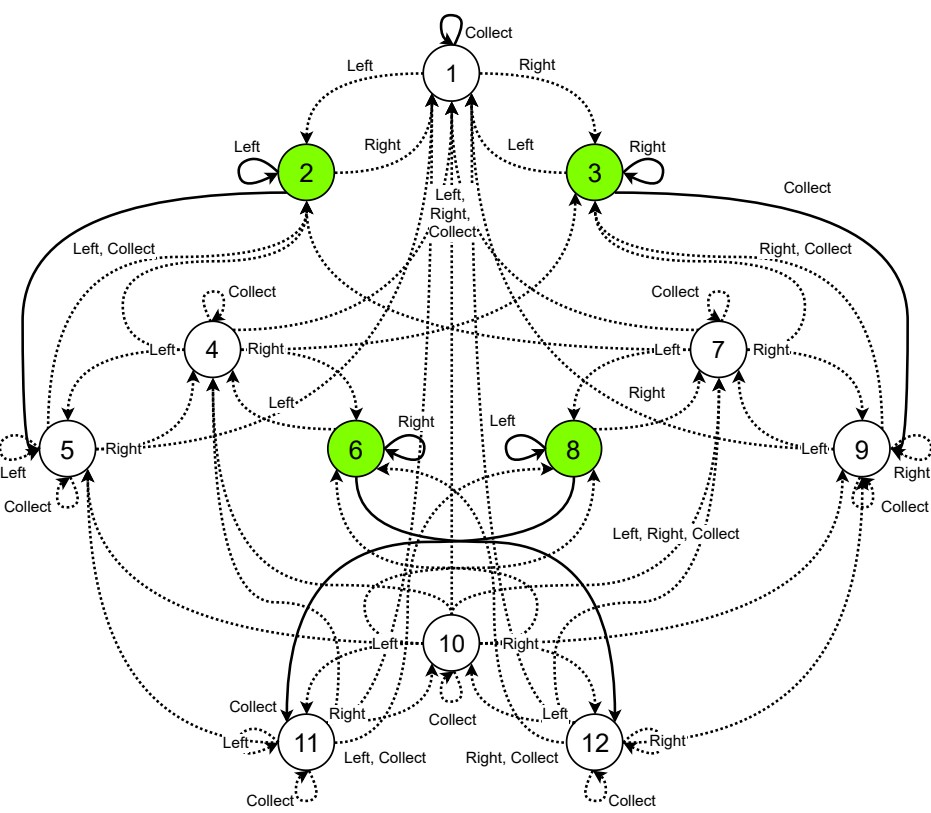

Figure 6: The Lobster Environment MDP. "Slippery" transitions at certain states are not pictured. Green nodes represent states where performing the `collect` action will yield a reward. Particular to only this figure, solid lines are deterministic transitions, whereas dotted lines are stochastic transitions.

"slips" and stays in the same location. At every time step, if a reward is not present, it regenerates according to its own Poisson processes, with an expected number of steps for regeneration of $\Lambda \doteq 10$ for each reward. We also visualize the full MDP as nodes and edges in Figure 6

## C.2 Hyperparameters and Experimental Setup

We now detail the experimental setup and hyperparameters swept for all agents mentioned in Section 3. We first detail all the shared settings and swept hyperparameters between all algorithms:

- Learning algorithm: $Sarsa(0)$
- Function approximator: $Linear$
- Optimizer: $Adam$
- Discount rate: $\gamma = 0.9$

- Environment train steps: 250K
- Max episode steps: 200
- Step sizes: $[10^{-2}, 10^{-3}, 10^{-4}, 10^{-5}]$

Because this environment is a continuing task, we evaluate our agents based on the undiscounted returns over 200 time steps, with no terminal time steps. All hyperparameter sweeps were done over 30 seeds, with the best hyperparameters decided for each algorithm based on mean undiscounted returns over these 200 time steps over these $250K$ steps and 30 seeds. The results reported are over 30 different additional seeds, with each additional seed run on the selected hyperparameters. We now briefly describe the two baselines that we employ as comparisons to the auxiliary input techniques we introduce, as well as minor details of the auxiliary input algorithms used in this section.

## C.3 Algorithmic Details and Additional Results

### C.3.1 Observations only

As a baseline for performance, we consider the observations-only agent. This agent simply uses the Sarsa(0) algorithm to learn a policy over the observations described in Equation 14. This agent is labelled in teal as "Observations".

### C.3.2 Value Iteration with Environment States

We also consider an optimal agent acting on the fully-observable version of this task, with full knowledge of the transition dynamics. We use this agent to see how close to optimal our partially observable agents can perform. We use the transition probabilities over the 12 possible states to perform value iteration (Bellman, 1957) to calculate the optimal value functions for control. We iterate through all states until our maximum change in value function over all states $\Delta$ is less than the threshold value $\theta = 10^{-10}$, $\Delta < \theta$. We use this optimal value function over the 12 states together with the transition probabilities to get the optimal policy. We evaluate this optimal policy over 200 steps and collect 1000 runs to get both the mean and standard error to the mean as the shaded dotted line in all learning rate plots.

### C.3.3 Exponential Decaying Trace

To use decaying traces as an auxiliary input for the Lobster environment, we take decaying traces of the elements in the observation vector that indicate whether or not each reward is collected. With the indexing and observations from Equation 14, our decaying trace for the Lobster environment is defined as:

$$\boldsymbol{m}^i_{t+1} \doteq \begin{cases} \lambda \boldsymbol{m}^i_{t-1} & \text{if reward } i \text{ is unobservable,} \\ \boldsymbol{o}_t[3i] & \text{otherwise.} \end{cases} \tag{15}$$

for each auxiliary input $i \in \{1, 2\}$ that corresponds to rewards in locations 1 and 2, respectively. Note that each auxiliary input here is a vector of size $\mathbb{R}^1$. As a reminder, $\boldsymbol{o}_t[3i]$ corresponds to the boolean that answers the question "Is the reward in location $i$ observable and missing?" Within our Lobster environment experiments, we use a decay rate of $\lambda = 0.9$.

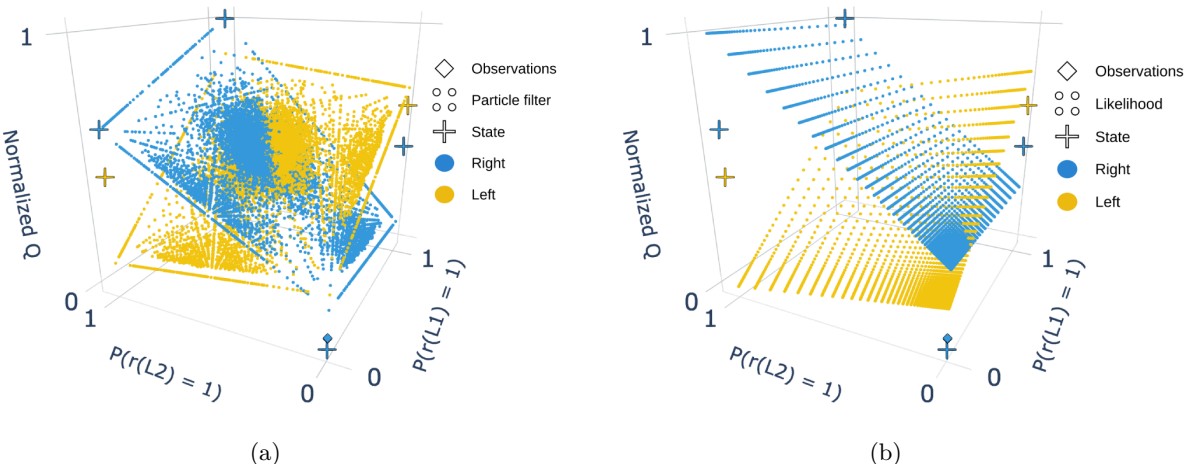

Figure 7: (a): The action-values for both `left` (yellow) and `right` (blue) actions for the value function learnt over particle filtering auxiliary inputs. Again, the observation-only agent is represented by overlapping diamonds in this plot. (b): The same plots for the action-values for the likelihood auxiliary inputs.

Put together, our auxiliary inputs are defined as $\boldsymbol{M}_t \doteq (\boldsymbol{m}_t^1, \boldsymbol{m}_t^2)$, which we use as part of our agent-state function. Our agent state for trace decay auxiliary inputs in the Lobster environment is the concatenation of the observation and the two auxiliary inputs in $\boldsymbol{M}_t$: $\boldsymbol{x}_t \doteq [\boldsymbol{o}_t, \boldsymbol{M}_t] \in \mathbb{R}^{11}$. The dimension of this agent state is the number of dimensions of the observation, plus two dimensions from the two auxiliary inputs $9 + 2 = 11$.

For the agents described in this section, we use the Sarsa (Rummery & Niranjan, 1994) algorithm to learn a control policy (with the exception of the ground-truth state agent, which uses value iteration to learn the optimal policy). We also use linear function approximation for all agents. For this trace decay agent, a step size of $\alpha \doteq 10^{-3}$ was selected from a hyperparameter sweep, with an epsilon of $\epsilon \doteq 0.1$ for the epsilon-greedy policy.

### C.3.4 Particle Filtering

With particle filtering, our approximate belief state becomes more accurate with more particles. In this environment, we instantiate the particle filter with 100 particles to begin with. In the rare case of particle depletion, where there are no particles in the current environment state, we reset all particle weights to be uniform and re-weight them from that time step onwards.

For this particle filtering approach in the Lobster environment, we approximate a distribution over the 12 underlying ground-truth states (as per Figure 6), based on the approximated distribution over state over $n_{particles} = 100$ particles. At every step, we follow the steps in Section 3.2.2 to get our approximate belief state as our auxiliary input. In this case, since we only have a single auxiliary input vector, we have that $\boldsymbol{M}_t \doteq \boldsymbol{m}_t$. In addition to this, our auxiliary inputs in this case define the entire agent state, since we already incorporate both observations and actions at each step in the particle filter update: $\boldsymbol{x}_t \doteq \boldsymbol{M}_t$. For this particle filtering agent, once again we leverage the Sarsa control algorithm as well as a selected step size of $\alpha \doteq 10^{-3}$ (from a hyperparameter sweep) and an epsilon of $\epsilon \doteq 0.1$.

We also show similar value function geometry plots to Fig. 2c, except for the belief distribution features in Fig. 7a. In this case, the bottom two $x$—$y$ axes correspond to, for each reward $i \in \{1, 2\}$:

$$P(r(Li) = 1) \doteq \sum_{s \in \mathcal{S}} \boldsymbol{m} \odot \mathbb{1}_{[s \text{ where reward } r(Li) = 1 \ \& \ s \text{ where the agent is in location } 0]},$$

which is the sum of the probabilities over environment states where the agent is in location 0 and the reward in location $i$ is present. In this particular visualization, we visualize all the features collected from multiple rollouts of the policy learnt by the particle filtering agent.

### C.3.5 Likelihood Predictions

To predict the ground-truth reward regeneration likelihoods, we assume we know the ground-truth rates of our Poisson processes for both rewards, which in this case is $\frac{1}{\Lambda} \doteq \frac{1}{10}$ for both rewards (on average, 1 reward regeneration every 10 steps). With this rate of reward, we calculate the likelihood of a reward being present.

We now describe how we model this future prediction for the Lobster environment as a likelihood. We do this through calculating, in closed form, the likelihood that r(L$i$) is present, given that some number of steps have elapsed with r(L$i$) as unobservable. In this auxiliary input scheme, those number of steps will depend on the number of steps since the agent has seen r(L$i$) missing *and* the average number of steps needed to reach L$i$. In this approach, we assume we have the privileged information of the rate $r$ at which either rewards are regenerated. Let $E^i_\tau$ be the event that the reward at location L$i$ is regenerated within $\tau$ steps, and $E^{i'}_\tau$ be the complementary event, where the reward does not regenerate after $\tau$ steps. Since this is a Poisson process, this means that the likelihood of a reward at location L$i$ regenerating after $\tau$ steps is:

$$P(E^i_\tau) = 1 - P(E^{i'}_\tau) = 1 - \exp\{\tau \cdot r\}, \tag{16}$$

where Equation (16) is simply the probability that at least one Poisson process occurs after $\tau$ steps. To calculate this prediction for our auxiliary inputs, we need the total number of steps in our trajectory between last observing r(L$i$) as missing, and the average number of steps to reach L$i$ from the current location. We find this total number of steps by summing these two number of steps, and finding the corresponding likelihood. Our auxiliary input is then defined as:

$$
\boldsymbol{m}^i_t \doteq 1 - \exp\left\{ \sum_{\tau=0}^{t} \mathbb{1}_{[(\boldsymbol{o}_\tau \text{ has r(L}i\text{) missing}) \ \& \ (\tau > \text{last observed r(L}i\text{) missing})]} \right.
$$
$$
\left. + \ \mathbb{E}_{\pi_i}\left[ \sum_{\tau=t+1}^{\infty} \mathbb{1}_{[\boldsymbol{o}_\tau[3i+2]=1]} \right] \right\}, \tag{17}
$$

where $\pi_i$ corresponds to the policy of going to location L$i$ (for L1 that is simply the policy that always goes left, for L2 this policy always goes right). $\boldsymbol{o}_\tau[3i+2]$ is the observation feature that answers the question "Is the reward at L$i$ unobservable?". Each auxiliary input defined here corresponds to a future prediction about the reward. In this case, our kernel function is simply a function which filters time steps based on if the step was in the past/present or future. Our preprocessing function is then defined by the function that returns the two element vector with the two predicates defined in Equation 17. Put together we get our likelihoods (one for each reward) which we use as auxiliary inputs for our agent.

Similar to our previous two auxiliary inputs, this approach also uses the Sarsa algorithm for control, with a step size of $\alpha \doteq 10^{-3}$ selected based on a hyperparameter sweep. The epsilon used here was also $\epsilon \doteq 0.1$.

### C.4 Results and Plotting Details

We now describe the axis and plotting details in Figures 2b and 2c. The $x$—$y$ axes (bottom two axes) capture two input features that represent the likelihood of each reward being present. For the ground-truth environment states, these features simply represent whether or not the rewards are present or not. Since we have two rewards in locations L1 and L2, this corresponds to four possible environment states at location L0: one for each possible state of the two rewards (since they can be either present or not present). For the observation-only agent, both of these corresponding features can only take on a single value due to partial observability: both features are 0 since at location L0 this agent will only ever see 0 elements for the features which correspond to whether or not the rewards in each location are there. We now describe how we plot the exponential decaying trace features.

From Equation 15, our exponential decaying traces *decrease* the more time elapsed since last observing each reward was missing. This means that our trace features for each reward should be inversely proportional to the likelihood of each reward being present. Given this, we plot the complement $(1 - \boldsymbol{m}^i)$ of each exponential decaying trace input in Figures 2b and 2c.

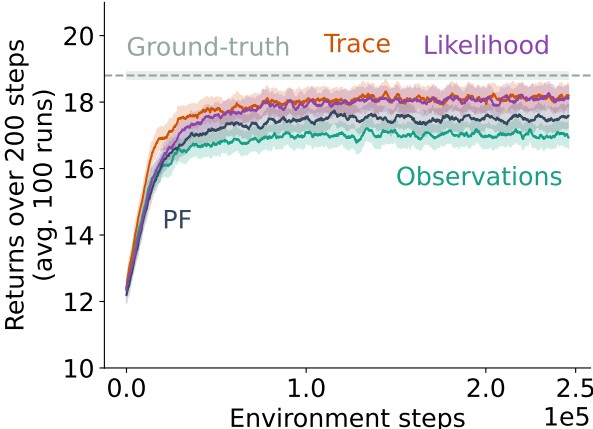

Figure 8: Results for a PPO agent on the Lobster environment.

### C.5 Stochastic Policies with Auxiliary Inputs

We show results for an agent that is able to learn stochastic policies in the Lobster environment in Figure 8. The agent utilizes the Proximal Policy Optimization (PPO, Schulman et al., 2017) algorithm, which is policy-gradient-based algorithm that explicitly learns policy parameters to represent stochastic policies. The observation-only agent here (in teal) performs significantly better than the Sarsa-based observation-only agent (also teal, in Figure 2a) due to the ability of the agent to learn a stochastic policy. The stochastic policy learnt by the observation-only PPO agent goes left *or* right at L0 with equal probability 0.5, allowing the agent to potentially seek out the reward that was not visited most recently, and is more likely to have regenerated. This stochastic policy achieves a much higher return as compared to the $\epsilon$-greedy policy learnt by the Sarsa algorithm, which will almost always dither between L0 and either L1 or L2, but not both. This means the agent essentially has to wait for the reward at either L1 or L2 to regenerate. These results also reflect the consensus that policy gradient algorithms generally perform better under partial observability than value-based algorithms—this is most likely due to the fact that stochastic policies allow an agent to perform better in partially observable environments.

From these results, the auxiliary input agents still outperforms the observation-only agent. While the increase may be more marginal as compared to the Sarsa algorithm, auxiliary inputs still increases performance over leveraging observations only. This implies that not all of the gains achieved by auxiliary inputs can be achieved by just switching to PPO or stochastic policies—there is additional information agents can utilize in auxiliary input features.

Results were run over 100 seeds as opposed to 30 seeds used in the other results in this paper due to the increased stochasticity from the stochastic policy. A $\lambda$ parameter of 0.95 was used, with $n$-step returns of $n = $ max. episode length used. A 1-layer neural network was used as the function approximator. Learning rates and hidden sizes were swept over 10 seeds, with values swept over $\{10^{-5}, 10^{-4}, 10^{-3}, 10^{-2}, 10^{-1}\}$ and $\{5, 10\}$ respectively. The 100 seeds were run over a learning rate of $10^{-2}$ and hidden size of 10 selected from the best performance (average reward over max. episode length) over the aforementioned sweep. No replay buffer was used for this agent, with the agent making an update after the roll-out of each episode.

## D Particle Filtering Environment and Experimental Details

### D.1 Environment Details

Below we consider all the details of the RockSample environment. We do not provide further details of the Modified Compass World environment here because the description in Section 4.1 together with Figure 10a fully specifies the environment.

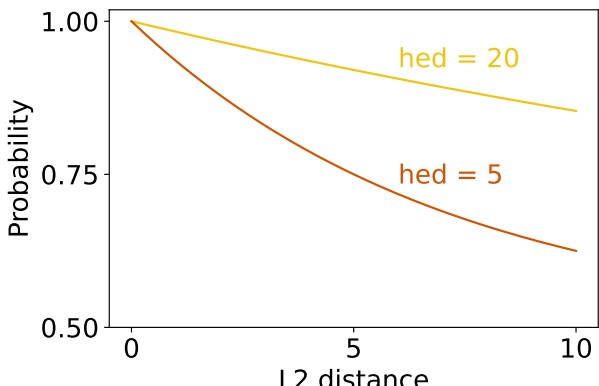

Figure 9: Half efficiency distance function plots: probability of a correct sensor reading as a function of distances for different half efficiency distances ($\delta_{hed} = 5, 20$).

### D.1.1 RockSample Environment Details

We now consider the environment and implementation details of the RockSample environment not considered in Section 4.1.

The position of each rock is determined in this environment based on a uniform sampling (without replacement) of 8 positions out of all possible $7 \times 7$ positions in the grid. This position is deterministically defined by the seed which the experiment is run on, and these positions do not change for each agent trained on a particular seed. The initial goodness and badness of rocks is determined based on a uniform Bernoulli distribution for every rock at every environment reset.

As mentioned when introducing this environment, the sensor available to the agent for checking the goodness and badness of rocks has noise proportional to the $L_2$ distance between the agent and the rock in question, which we denote as $\delta$. This noise is essentially based on another Bernoulli distribution, where with probability $p(\delta)$ the sensor returns the correct sensor reading of the rock, and with probability $1 - p(\delta)$ the agent receives an incorrect reading. The probability function $p$ is defined by the *half efficiency distance* $\delta_{hed}$ of the sensor. Overall, this probability function is defined by the function:

$$p(\delta) \doteq 0.5 \times (1 + 2^{-\frac{\delta}{\delta_{hed}}}). \tag{18}$$

We plot this function in Figure 9. We also discuss results for different agents introduced in Section 4 in RockSample for different half efficiency distances in Section D.3. In our results in Section 4.2, we use a half efficiency distance of 5.

## D.2 Environment-Specific Algorithmic Details and Hyperparameters

We now detail the environment-specific algorithmic details and hyperparameters swept for all agents pertaining to Modified Compass World and RockSample.

### D.2.1 Modified Compass World Experimental Setup and Hyperparameters

For our Modified Compass World experiments, we use and sweep the following hyperparameters:

- Learning algorithm: $Sarsa(0)$
- Function approximator: Neural Network
- Layers: 1
- Hidden units: 100
- Optimizer: *Adam*

- Discount rate: $\gamma = 0.9$
- Environment train steps: 1M
- Max episode steps: 1000
- Step sizes: $[10^{-3}, 10^{-4}, 10^{-5}]$
- Number of particles: 1 for each possible start state ($9 \times 9 \times 4 - 1 = 323$)

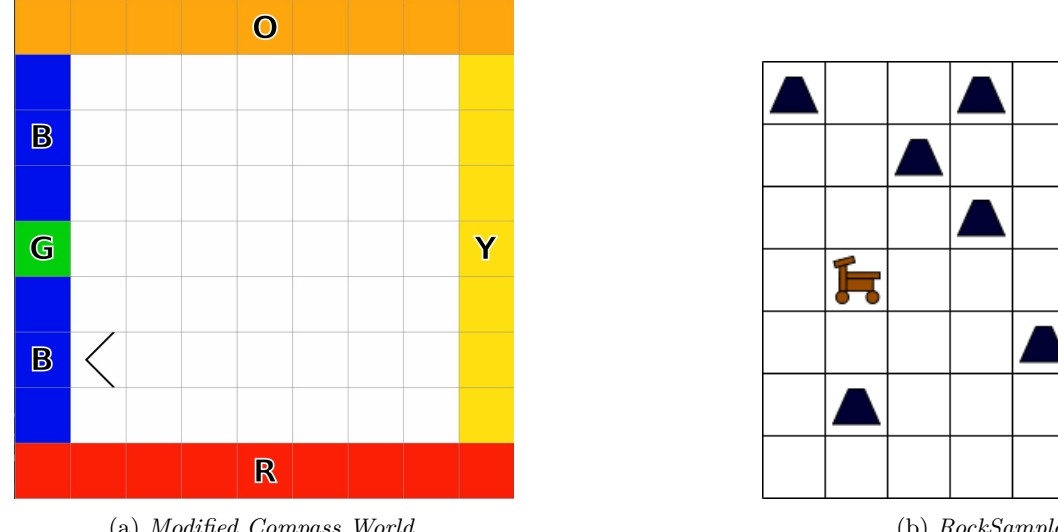

(a) *Modified Compass World*

(b) *RockSample(7, 8)*

Figure 10: (a) Modified Compass World and (b) RockSample(7, 8) environments for evaluating approximate belief states as auxiliary inputs.

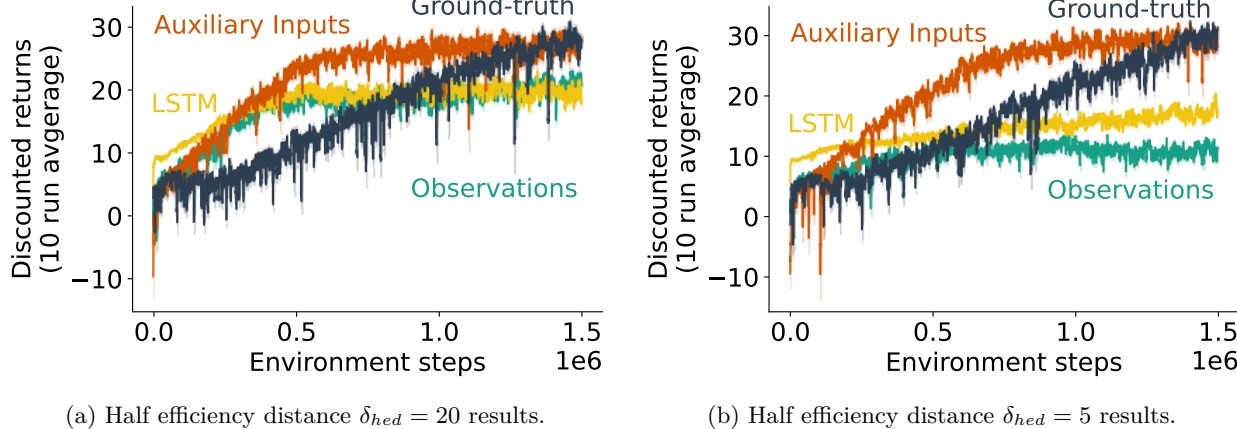

(a) Half efficiency distance $\delta_{hed} = 20$ results.

(b) Half efficiency distance $\delta_{hed} = 5$ results.

Figure 11: RockSample results for both $\delta_{hed} = 5, 20$. In (a) and (b) mean and standard error to the mean are shown over 10 seeds. We use $\delta_{hed} \doteq 5$ for the results in our work.

All hyperparameter sweeps were done over 10 seeds, with the best hyperparameters decided for each algorithm based on mean discounted returns over these 200 time steps and 10 seeds. We then use these selected hyperparameters and run experiments for 30 different seeds to obtain the results presented in Figure 3a. For the LSTM agents in this environment, the agents all use one-hot action concatenation (Schlegel et al., 2021) with its input features to the LSTM cell. An ablation study for this action concatenation is done in Appendix D.3.2.

**Ground-truth Agent** In Modified Compass World, the ground-truth agent converges to a much higher return than the other agents. This is because it is fully observable and has more information available to it in its data stream than the other agents—namely the position and orientation of the agent. The learnt policy for this agent traverses directly to the green goal state, without having to localize first.

**Observations Only** With the observations-only baseline, our observation vector is defined by a vector of size 5, where each feature corresponds to whether or not the color directly in front of the agent is being observed. An all zero vector represents no color being shown in front of the agent.

**Particle Filtering** We use particle filtering to approximate a belief state over the possible pose and position of the agent. Our feature vector for this approach is of size $7 \times 7 \times 4$, where these features represent all possible combinations of positions and poses of the agent. This position and pose belief state is approximated through the particle filtering approach described in Section 3.2.2, where emission probabilities are simply binary variables representing whether or not each position and pose combination can emit the given color. For this environment, we use one particle for each possible starting position and pose combination, $7 \times 7 \times 4$, since the environment dynamics are deterministic outside of the initial start state.

**Recurrent Neural Network** Finally, our RNN-based approach uses the same observations as described in Appendix D.2.1, except with an LSTM as the function approximator. In addition to using a recurrent neural network for function approximation, we also use *action conditioning* (Schlegel et al., 2021). In our setting, actioning conditioning simply consists of concatenating a one-hot encoding of the previous time step's action to the observation vector fed into the RNN. As a point of clarification, the LSTM-based agent conditions on both the observation and action at every time step.

### D.2.2 RockSample(7, 8) Experimental Setup and Hyperparameters

In our RockSample(7, 8) experiments, we leverage a replay buffer (Lin, 1992) for all of our experience. We use and/or sweep the following hyperparameters:

- Learning algorithm: $Sarsa(0)$
- Function approximator: Neural Network
- Layers: 1
- Hidden units: 100
- Optimizer: *Adam*
- Discount rate: $\gamma = 0.99$

- Environment train steps: 1.5M
- Max episode steps: 1000
- Step sizes: $[10^{-3}, 10^{-4}, 10^{-5}]$
- Buffer size: $[10K, 100K]$
- Number of particles: 100

All hyperparameter sweeps were done over 10 seeds, with the best hyperparameters decided for each algorithm based on mean discounted returns over these 200 time steps. We then use these selected hyperparameters and run experiments for 30 different seeds to obtain the results presented in Figure 3b. For the LSTM agents in this environment, the agents also use one-hot action concatenation (c.f. Appendix D.3.2).

**Ground-truth Agent** For the RockSample ground-truth agent, all the actual moralities (goodness or badness) of the rock are available to the ground-truth agent at the start of an episode. The learnt policy for this agent traverses directly to the good rocks, collects them, then exits to the right.

**Observations Only** With the observations-only baseline, our observation vector is defined by a vector of size $7 + 7 + 8 = 22$, where the first $7 + 7$ features represent one-hot encodings of the $x$ and $y$ coordinates of the agent respectively, and the final 8 features represent the most recently observed rock moralities (goodness or badness). In our implementation of RockSample, these observed rock moralities are initialized at 0.5, and take on values depending on the most recent check of each rock. So if rock number 1 was checked and seen as good 5 steps ago, and this was the most recent check of rock 1, then the feature representing this rock would be a 1 feature, representing a good rock.

**Particle Filtering** To approximate a belief distribution over state, we leverage particle filtering as mentioned in Section 3.2.2. In this approach, our input feature vector is also defined as a vector of length $7 + 7 + 8$. The first $7 + 7$ features are once again a one-hot encoding of the $x$ and $y$ coordinates for the first and second 7 features respectively. The final 8 features are an approximate belief state of the current state of the rock moralities, instead of the underlying state of the environment. This is to reduce the dimensionality of the input features. These last 8 features are simply the normalized sum over the weights over all particles. For this particle filtering algorithm, we start with 100 particles as well.

**Recurrent Neural Network** Finally, our RNN-based approach uses the same observations as Appendix D.2.2, except with an LSTM as the function approximator. Similarly to the Modified Compass World LSTM agent, we also use action conditioning here.

Since we use a replay buffer with LSTMs for this algorithm, to fix a truncation length for T-BPTT, we sample trajectories from our replay buffer of length *truncation length*, and roll our trajectories out and propagate gradients backwards across the sampled trajectory. This requires us to store and sample hidden states from our LSTM in our replay buffer. Similarly to the Compass World LSTM agent, this agent also conditions on both the observation and action at every time step.

### D.3  Ablation Studies

Here we list the ablation studies we performed over both our environments and select algorithms.

#### D.3.1  RockSample(7, 8) Half Efficiency Distance Experiment

We perform a small experiment to see the effect of the half efficiency distance $\delta_{hed}$ on the performance of our algorithms in Figures 11a and 11b. From these learning curves we can conclude that a lower $\delta_{hed}$ (or a less accurate sensor over distance to rocks) does not significantly affect the performance of the particle filtering auxiliary input, nor does it affect the performance of the ground-truth agent. This parameter seems to affect the LSTM-based agent and observations-only-based agent the most, decreasing performance for both. The results presented in the main body of this work use $\delta_{hed} \doteq 5$.

#### D.3.2  LSTM Action Concatenation Ablation

We conduct another small ablation study on action concatenation with the LSTM agent on both Modified Compass World and RockSample(7, 8). Results are shown in Figure 12. Action conditioning seems to generally help the LSTM agent in both environments—this is because we are conditioning on and providing more information than the LSTM agent not conditioned on actions. We note that the degree in which this action conditioning helps varies from environment to environment. In Modified Compass World, action conditioning is vital to the performance of the LSTM agent, whereas in RockSample(7, 8) we observe only potential marginal performance increases (potential since we have overlapping standard error bars). This reveals a further point with regards to action conditioning for RNNs: actioning conditioning will help depending on how much knowledge of the action resolves partial observability. In Modified Compass World, where the state is extremely partially observable, knowing the previous action resolves a big portion of the partial observability of the environment, whereas in RockSample, the previous action does not reveal too much about the environment state.

## E  Fishing Environment and Experimental Details

### E.1  Fishing Environment Details

In this section, we describe the specifics for both Fishing environments, as well as the environment parameters for both Fishing 1 and 2, including the rates of change for currents and the rates of regeneration for rewards.

There are 4 actions in both environments, each corresponding to moving in one of the cardinal directions. Both environments are represented by an $11 \times 11$ image with 4 locations that emit non-zero rewards, as denoted by the green points. Like in the Lobster environment (c.f. Section 3.1), in these environments the four rewards regenerate stochastically after being collected. Once collected, the rewards stay and do not disappear without being collected, and the agent receives a reward of $+1$. Currents in these environments also change stochastically over time. The multi-directional arrows represent these shifting currents for each given position, and the directions in which they might be in. In addition to these sources of stochasticity, the agent also has a chance of "slipping" at every time step and remaining in the same position. Beyond this, there are also walls throughout the environment, denoted by the dark blue tiles, that block the agent from traversing to or through. Bumping into a wall results in a no-op. Finally, Fishing 2 in Figure 4c also has a glass wall denoted by the blue tiles, which the agent can see through, but cannot traverse through. We now

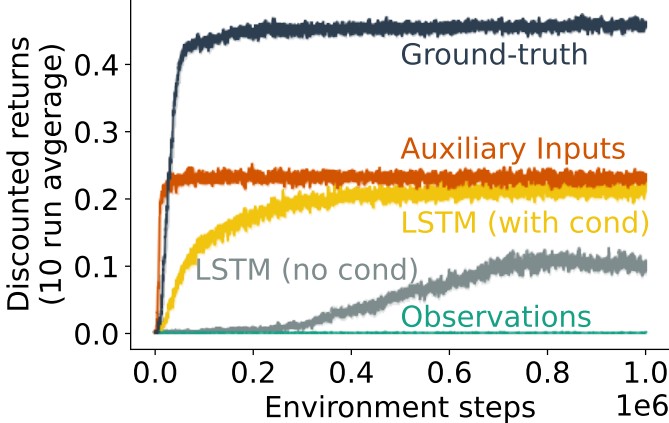

(a) Modified Compass World LSTM action conditioning ablation.

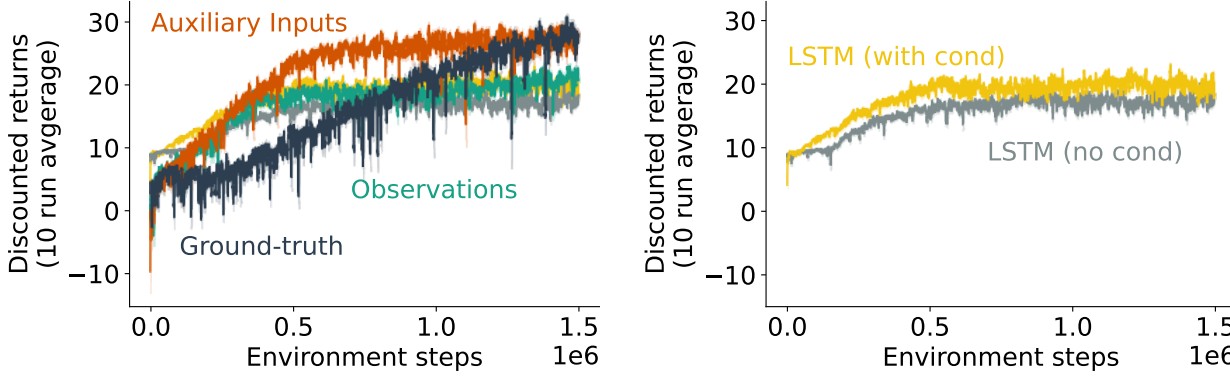

(b) RockSample(7, 8) LSTM action conditioning ablation.

(c) Fig. (b), but comparing only action conditioning vs no action conditioning.

Figure 12: Action conditioning ablation for LSTM. Results are over 10 seeds for both Modified Compass World and RockSample(7, 8).

describe the observations that the agent sees at every step. As for starting position, the agent starts in the $(x, y)$ position $(5, 5)$ in both environments.

### E.1.1 Mapping and Observations

One part of the partial observability of this environment is the limited observation an agent receives at every step. The agent receives a $5 \times 5$ agent-centric observation vector at every step. All agents view an accumulated (over time) agent-centric *map* of these observation vectors, which we call the agent map. For our $11 \times 11$ grid world, this amounts to a square observation of length $11 + 11 - 1 = 21$—the dimensions beyond the length 11 of the grid world account for the agent-centric view when the agent is at the edges of the grid world. At every step, the $5 \times 5$ observable area is updated on the agent map with the given coordinates of the agent. Hence, with every additional step, since rewards and currents change stochastically over time, previously observed (but currently unobserved) missing rewards and currents on the agent map are less and less accurate as time progresses.

The agent's view is also *obstructed* by the walls in the environment. If an agent is next to a wall, then everything behind the wall is obstructed, even if the area was meant to be observable. This is true for all walls except for glass obstacles, denoted in blue in Fishing 2. These glass obstacles act as walls, except that the agent is able to see through them, unlike normal walls. Glass walls are placed here so that the agent is able to see the direction that the currents are facing within the tunnel to the reward.

The agent observes a tensor of shape $21 \times 21 \times 6$, where the last dimension indicates the channels for different aspects of the environment/observation. The agent is able to see obstacles (walls and glass walls, 1st channel), currents and their direction (next 4 channels), and finally the locations of rewards if they are present (last channel).

### E.1.2 Stochasticity in the Environment

Both rewards and currents in both environments are defined by Poisson processes, with potentially different rates corresponding to each reward and each current. We now describe these rates for both environments. All currents depicted with a single-direction arrow denote a current that is static and does not change direction. Note for all currents, when a current is sampled to change, we sample uniformly at random from the remaining current directions, excluding the original current direction.

Besides the stochasticity from the Poisson processes, the agent also has a 0.1 probability of "slipping" for a move, when the agent takes an action, there is a probability 0.1 the agent simply stays in place.

**Fishing 1 Poisson Processes** In Fishing 1, all our stochastic processes have equal rates of regeneration or current flipping. This rate is 60—or on average, these Poisson processes will activate in expectation over after 60 steps. This larger rate is such that the agent is more incentivized to go collect other rewards, rather than staying at one particular reward and waiting for regeneration.

**Fishing 2 Poisson Processes** In Fishing 2, rates of reward generation are all set to 50, except for $(y, x)$ coordinates $(8, 9)$, which has a rate of regeneration of 100. Note, from here onwards we will list positions as tuples of coordinates of $(x, y)$. As for our currents, we group our currents based on their reward regeneration rates (in order from left to right, top to bottom in the grid world):

10: $(0, 6), (0, 7), (2, 5), (2, 6), (4, 1), (5, 7), (5, 8), (5, 10), (8, 7), (9, 7)$.
20: $(1, 3), (1, 4), (7, 0), (7, 1), (8, 5), (9, 5), (10, 2), (10, 3), (10, 4)$.
30: $(5, 2), (5, 3), (5, 4)$.
40: $(0, 2), (2, 0), (2, 1), (3, 9), (3, 10), (6, 2), (6, 3), (6, 4), (9, 8), (10, 8)$.

## E.2 Fishing-Specific Algorithmic Details and Hyperparameters

We now detail the algorithmic details and hyperparameters swept for all agents on the Fishing environments. In both our Fishing experiments, we leverage a replay buffer (Lin, 1992) for training. We list the hyperparameters swept for all our algorithms below:

For the Fishing experiments, we use a convolutional neural network to parse our agent map tensor. We use and/or sweep the following hyperparameters:

- Learning algorithm: $Sarsa(0)$
- Function approximation: Convolutional Neural Network
- Layers: 1
- Hidden size: 64
- Batch size: 64
- Optimizer: $Adam$

- Discount rate: $\gamma = 0.99$
- Environment train steps: 2M for Fishing 1, 12M for Fishing 2
- Max episode steps: 1000
- Step sizes: $[10^{-4}, 10^{-5}, 10^{-6}, 10^{-7}]$
- Buffer size: $100K$
- Evaluation frequency: 2K for Fishing 1, 10K for Fishing 2

All hyperparameter sweeps were done over 5 seeds, with the best hyperparameters decided for each algorithm based on mean discounted returns over the last 100 evaluation steps. Results in this section use *offline evaluation* returns over environment steps. Offline evaluations are conducted every *evaluation frequency* steps (as listed above). We run 5 test episodes per offline evaluation, and also average over these test episodes as well as seeds for a final average return for a given evaluation step at a certain training step. We then use these selected hyperparameters and this offline evaluation to run experiments for 30 different seeds to obtain the results presented in Figures 4b and 4d.

### E.2.1 Convolutional Neural Network Architecture

We now detail the architecture for our convolutional neural network. All our convolutional layers use a stride of 1 and no padding:

- Conv2D(output channels = 32, kernel size = 10)
- Relu activation
- Conv2D(output channels = *hidden size*, kernel size = 7)
- Relu activation
- Conv2D(output channels = *hidden size*, kernel size = 1)
- Linear layer with output $n_{actions}$

### E.2.2 Convolutional Neural Network LSTM Architecture

Our LSTM implementation is a convolutional neural network with an LSTM layer after the convolutional layers:

- Conv2D(output channels = 32, kernel size = 10)
- Relu activation
- Conv2D(output channels = *hidden size*, kernel size = 7)
- Relu activation
- Conv2D(output channels = *hidden size*, kernel size = 1)
- Relu activation
- Linear layer with output *hidden size*
- LSTM(hidden state size = *hidden size*)
- Linear layer with output $n_{actions}$

### E.2.3 Exponential Trace Implementation Details

While we describe the approach to using exponential decaying traces for the Fishing environment in Section 5.2, we go into detail here with regards to implementation details and hyperparameters swept.

Our exponential decaying traces are simply another channel in our agent map tensor, with the same size of $15 \times 15$. It is a tensor of elements in the range of $[0, 1]$, with each element denoting how long it has been

since observing that particular position (where 1 denotes the agent is currently observing this area, and 0 denoting it has never observed this position).

As for the decay rates, we swept the following rates: $[1, 0.95, 0.85, 0.65]$.

### E.2.4 Recurrent Neural Network Implementation Details

With our RNN implementation, we simply use the same technique of training an LSTM with a replay buffer as in Appendix D.2.2, where we sample trajectories of length *truncation length* for T-BPTT. We swept the following truncation lengths for both Fishing environment hyperparameter sweeps: $[1, 5, 10]$.

