# OpenReview forum: "Agent-State Construction with Auxiliary Inputs"
_TMLR — Accepted by TMLR_

### Review · Reviewer_pWPv · 2023-01-10

**Summary Of Contributions:**

The manuscript presents a framework for describing and categorizing techniques to regress information about the underlying state of the environment in partially observable RL settings in a model-free fashion (i.e. at the agent input level). The manuscript presents a short survey of techniques for augmenting standard agent observations to tackle temporal and dynamical partial observability.

The authors analyze a few classes of techniques by picking common but representative instantiations of such techniques (decaying traces, particle filters, GVs) to represent cases where such input augmentations may contain -- potentially learnt / estimated -- data about the past, present, or the future wrt. a policy rollout in an environment instance. They evaluate these in a few low dimensional toy environments.


**Audience:**

Yes

**Claims And Evidence:**

Yes

**Requested Changes:**

I think the manuscript would be particularly improved by:

1. Drastically simplifying the description of in the main manuscript, moving some of the verbosity (that to be fair does aid correctness) to the appendix.  Further improvement could also include adding visualizations of the representation of the state as used in some of the more realistic environments for each of the techniques studied.

2. Questioning whether the claim that trace features can work generally work as well or better than recurrent models is sound given the complexity of the presented empirical settings. Would this claim hold with higher dimensional (or even just somewhat larger) input spaces. FWIW I believe the claim to be correct, but that's my own personal bias, rather than being something that I can gather directly from the manuscript as-is :-) Thus, I would consider attempting to compare these methods in "harder" (but standard) environments, to strengthen this part.

3. Attempting to use the framework to learn _more_ about these methods, rather than simply reformalizing them into -- what might appear to the inexperienced reader  -- unnecessarily complex descriptions. As it stands, I don't understand the point of the evaluation setting. What are the authors attempting to communicate to the readers? Can the implied expectations (and payoff!) be made clearer in writing? What do we gain from using the framework instead of just hacking away? (Perhaps types of information gain?)


**Strengths And Weaknesses:**

## Strengths

1. I believe the fundamental framework to be both correct and useful in its generality. It seems to describe fairly well all input augmentation systems I could come up with and recall whilst reading the manuscript. Thus I think it is a valuable contribution, and I welcome efforts to more formally describe some of the commonly-thought-to-be-"hacky" steps we apply to our environment-agent problems as RL researchers when attempting to lower the cost of partial observability.

2. The manuscript is well thorough in delineating constraints and structure in the framework, and makes a reasonable case for the presented evaluation settings. It is clear that the authors are attempting to develop an empirical argument bottom up using the presented toy problems such that later work might attempt to reproduce relatively simple (to describe, at least) dynamics in more complex environment-agent tuples.

3. Overall __I__ found the paper and its arguments relatively clear, precise, and not surprising (which IMO in this case is a good property). I reckon an expert would probably not struggle to reproduce the results.

## Weaknesses

1. The paper is not an easy read, which is a shame considering the relatively straightforward contributions and findings. The main culprit seems to be the excessive use of formalism in key sections (2 and 3), which I don't believe strikes the right balance between correctness and clarity; but the later sections that serve to evaluate the framework using some common algorithmic instances are also complex in such a way that makes it hard to map my mental model of the discussed common techniques (which I hope is relatively representative).

2. I found it strange how (a) the manuscript presents the framework as general wrt. (partially observable) agent-environment tuples, but (b) constrains the evaluation to a relatively small model-free agent class (value-based), and as such single agent (SARSA) within it. Given that the experiments seem to be relatively cheap, and that the manuscript's primary narrative seems to be one of analysis, I was expecting a broader empirical discussion, and generally found these choices to make for underwhelming data.

3. Follow up from (2), the experimental setup is just too toy-ish. I strongly believe that use of toy / constrained environments is reasonable and absolute fair game when attempting to exactly study and demonstrate properties, tradeoffs, and emergent behavior of proposed methods in an empirical fashion. However, this particular topic is interesting to analyze and formalize primarily because it tackles the "hacky" methods that we as a community kinda winged usage across wildly diverse domains and agents, diverse along a large set of dimensions (complexity, observation types, stochasticity, structure, etc). The lack of empirical results in at least slight-above-toy domains (e.g. ALE) was surprising, and from my perspective reduced the impact of the empirical comparison of types of auxiliary inputs.

4. I think the empirircal results are not really novel. Plenty of previous work has used these auxiliary inputs to effectively make popular problems tractable from an observability perspective, so the results don't seem to be new in any meaningful way. What new information are they trying to communicate?

---

> ### Author Response · Authors · 2023-02-03
> **Re: Review of Paper599 by Reviewer pWPv pt.1**
>
> Thank you for the support and feedback.
>
> The biggest concern we’d like to address right off the bat is with “where the novelty lies” in our work.
> In most use-cases of reinforcement learning, practitioners do all kinds of feature engineering, especially in the most recent big success stories of deep reinforcement learning. This paper focuses on one of the most pervasive kinds of feature engineering: augmenting the inputs to encode different summaries to mitigate partial observability. This is pervasive because almost all interesting applications are partially observable. We provide a simple unification of these seemingly heuristic successes and then provide 3 case studies that provide empirical insight into when you should use such approaches and also why they might help. We’ve further clarified these points in the introduction (Section 1, blue text).
>
> On top of this, we’ll try to be thorough in addressing the requested changes you mentioned below:
>
> **1. “Drastically simplifying the description of in the main manuscript”**
>
> We’ve made quite a few major edits to the manuscript to address the lack of clarity in our formalism. As per your suggestion, we have simplified the formulation of the auxiliary input (Sections 2 and 3, in blue text). Our new simpler formalization of auxiliary inputs is now a general function over the entire trajectory of the agent. In addition to this, we have relegated the additional details of the convolutional approach to auxiliary inputs to the appendix (Appendix B).
>
> **2. “Questioning whether the claim that trace features can work generally work as well or better than recurrent models is sound given the complexity of the presented empirical settings.”**
>
> This is a great point. We agree that this would potentially be the case generally, but we’re also careful in not making this claim in our paper, and have never meant to imply this either. There are many cases where an exponential decaying trace of features work better than RNNs, one of those cases being the environments presented in our study. There are also many cases where RNNs can do better than traces. This is why we presented results for the combination of the two approaches, and showed that this approach actually works better than any individual approach, instead of trying to solely compare the two in this domain. As for comparing the two (LSTMs and exponential decaying trace features) on other (potentially larger) domains, although we definitely agree that this would be an interesting result that merits further study, it would transform the main argument of the section (auxiliary inputs can be scaled, and incorporated with RNNs) into comparing LSTMs with exponential decaying traces as features. We believe that this comparison, as well as other algorithmic and environmental comparisons (such as control algorithm, type of partial observability etc.) would merit an independent work of its own, and have added this to our future works section (again, in blue in Section 6).
>
> Continues in pt. 2 below

---

> > ### Author Response · Authors · 2023-02-03
> > **Re: Review of Paper599 by Reviewer pWPv pt.2**
> >
> > Continues from pt. 1.
> >
> > **3. “Attempting to use the framework to learn more about these methods, rather than simply reformalizing them into -- what might appear to the inexperienced reader -- unnecessarily complex descriptions. As it stands, I don't understand the point of the evaluation setting. What are the authors attempting to communicate to the readers? Can the implied expectations (and payoff!) be made clearer in writing? What do we gain from using the framework instead of just hacking away? (Perhaps types of information gain?)”**
> >
> > Thank you for the suggestions in these questions.
> > The argument behind the work isn’t to solely use the framework itself to develop auxiliary inputs, but to use this framework to try and make more informed decisions when a practitioner is deciding on auxiliary inputs for de-aliasing. This idea of leveraging our formalism to create auxiliary inputs is in itself a very interesting research question, which we believe is out of scope for this work, and has been added to the future works section.
> >
> > As for our evaluation setting, the point of our work isn’t to show that approach X outperforms algorithm Y - instead we are looking to understand why the “hacky” methods that our community has been employing for Z amount of years works across so many different domains. Section 3 does this through our visualizations of all three auxiliary input approaches, and comes to the conclusion that auxiliary input expands the input space to our value function to better discriminate between otherwise aliased states - a point that might not be clear if we instead compared N algorithms and auxiliary inputs on M benchmark tasks. Instead we are looking to:
> > Provide a unifying framework to encapsulate what auxiliary inputs are, and
> > Give practitioners advice on feature engineering and what types of feature engineering may be helpful for neural networks in different partially observable domains.
> >
> > To help improve clarity in terms of what we’re trying to communicate to the reader, we’ve made changes to the paper such that we now start each experimental section with the purpose behind the experiments, and conclude the sections with the lessons learnt from these experiments (specifically for Sections 3, 4, and 5).
> >
> > Finally, the reason we have decided to evaluate different auxiliary inputs on a single control algorithm is because the main thesis of this work focuses on how auxiliary inputs de-alias states for better value-function learning, and allows learning agents to represent a better policy. This is in itself a general result that doesn’t depend on the control algorithm used on top of the value learning. In order for us to understand the need for additional RL algorithms a more, we would like to clarify a few things with the reviewer and pose a few questions we should ask. Namely, what additional base RL-methods might we include, and what could we learn from such experiments. What potential hypothesis could we make here? Here is a selection of representative algorithms we could test:
> >
> > 1. policy gradient methods such as PPO or SAC
> > 2. Model-based approaches like Dreamer
> > 3. Asynchronous learning systems like A3C
> >
> > For 1, we should not expect policy gradient methods to somehow be able to learn in partially observable domains with a standard feedforward architecture. Indeed, the need for memory is agnostic to the choice of value-based method vs policy gradient method.
> >
> > In 2, for model-based RL, the agent learns both a policy and a world model. It is well-known that a major challenge in model-based RL is inaccurate models causing rollouts to produce invalid states and bad updates (Talivitie, 2016). This problem is only amplified more in partially observable domains; improving the representation used by the model via auxiliary tasks would be very promising indeed, but a massive and tangential quest when progress in MBRL has been, up until very recently, plagued with performance issues that are well beyond the scope of this paper.
> >
> > Finally, asynchronous methods with distributed actors still require good state information. Much of the work in this area is in fully observable deterministic domains like Mujoco. There certainly could be improvements from auxiliary tasks there and it would be nice to show. However, we are adhering to a simple principle here: demonstrate the phenomenon and benefits clearly in a setting where we can reduce confounding factors and understand where the performance improvements are coming from. Adding more algorithms would add significant complexity for unclear gain.
> >
> > If the concern is that the benefit demonstrated here would not hold for other base RL algorithms we do not understand why this would be the case, and invite the reviewer to expand on the point.

---

### Review · Reviewer_Vzj8 · 2023-01-10

**Summary Of Contributions:**

This paper gives a new formalism on input augmentation in partially observable sequential decision making problems. They provide a set of rules to construct augmentation from interaction history, under which several common practices of augmenting the input or observations can be viewed as special case. They provide empirical evaluation in two finite-state and a larger pixel-based environment to show the efficacy of these approach.

**Audience:**

Yes

**Claims And Evidence:**

Yes

**Requested Changes:**

Regarding the TMLR's evaluation criteria, the weakness point 1, 2 and 4 need to be clarified in the paper.

**Strengths And Weaknesses:**

Strengths:

1. The problem of how to augment the observation history is important as most real-world decision problems are either partially observable, or the observation dimension is too high to be efficiently modeled by the agent. Heuristic methods used in the augmentation are ad-hoc and lack principled analysis. This paper provides a unified formalism of input augmentation in this setting.
2. This paper discussed the relationship between the proposed framework and many classical methods to solve partial observability.

Weaknesses:

1. As a new formalism, some definitions in the proposed framework is not clearly or concise enough.
 - 1.1 Is auxiliary input defined at the interface between agent and environment or it is a computation module inside of the agent. What is the difference between auxiliary input and observation and why one cannot be viewed as (part of) the other.
 - 1.2 It brings unnecessary notations and levels of concept to define $M_t$ as N different auxiliary input function $m_t^i$. Given each $m_t^i$ is already a vector function with m dimension, it is mathematically equally representative as the definition of $M_t$.
 - 1.3 It says "we define auxiliary inputs to summarize the past, present and/or future.". It need to be explained more how to summarize the future and compute the auxiliary inputs in the test time.
2. In section 2 the authors define agent-state function as a recurrent function. In section 3, it says "we investigate the use of auxiliary inputs as an input into the agent-state function." However, in later parts from section 3.1 and especially section 3.3, the authors study the use of  auxiliary inputs or observations as an input directly to the state-action value function. Please clarify that either my understanding is wrong or what's the relationship between the proposed framework and the recurrent agent-state function.
3. The larger scale experiment in fishing domains only shows one form of auxiliary input.
4. The experiment in fishing domains can be stronger by adding more discussion. There could be more discussion on the choice of preprocessing function and why LSTM + Exp Trace and LSTM learns differently. (LSTM with observation as input has the capacity to model a exponentially weighted summation of $g(O_t)$)

Minor clarity issue:
 - Technically the auxiliary input is not actually another input which provide new information, but an aggregation function of observation-action history. It seems confusing to me why name it as input rather than aggregation or augmentation function.

---

> ### Author Response · Authors · 2023-02-03
> **Re: Review of Paper599 by Reviewer Vzj8**
>
> Thank you for the kind words and feedback. We have considered points 1, 2 and 4, and have edited our manuscript accordingly. Please see our response to each point of feedback below, and let us know what you think:
>
> **1.1 “Is auxiliary input defined at the interface between agent and environment or it is a computation module inside of the agent”**
>
> This will depend on your definition of the agent-environment boundary. We draw this boundary as any processing of the observations/rewards/actions as being internal to the agent, whereas the environment simply emits observations and rewards based on committed actions. Currently we define auxiliary inputs as something inside of the agent that processes observations and actions at every step, and fed into the agent-state function as an additional input for decision making. We’ve included a visualization of this in
>
> **1.2 Unneccessary notation**
>
> As per your suggestion, we have simplified the formulation of the auxiliary input (Sections 2 and 3, in blue text). Our new simpler formalization of auxiliary inputs is simply a general function over the entire trajectory of the agent. In addition to this, we have relegated the additional details of the convolutional approach to auxiliary inputs to the appendix (Appendix A).
>
> **1.3 “It need to be explained more how to summarize the future and compute the auxiliary inputs in the test time.”**
>
> Thank you for pointing out this ambiguity. Section 3.2.3 gives one example of how to summarize the future and compute auxiliary inputs. We also mention in our future works section potential avenues for summarizing the future for auxiliary inputs - techniques such as PSRs or GVFs are both promising candidates that we want to consider in future work. As for how to summarize the future when testing the agent, our formalism makes no assumptions about differences between training and test time - all the information available for auxiliary inputs would be available to the agent at both train and test time.
>
> **2. “Please clarify what's the relationship between the proposed framework and the recurrent agent-state function”**
>
> Thanks for bringing this up - we previously defined the agent-state function as a recurrent function, but proceed to not use recurrency in any of our studies. We have further clarified this in an update to the manuscript (Section 3, page 4, blue text). Specifically, we note that while the recurrent formulation of the agent-state function is the most general case, most of the time in practical scenarios augmenting inputs with auxiliary inputs provides enough state information for the problem at hand, and is easier/more stable to train/works better in general.
>
>
> **4. “There could be more discussion on the choice of preprocessing function and why LSTM + Exp Trace and LSTM learns differently. ”**
>
> This is a great suggestion. We have changed the discussion portion in Section 5 (blue text) to reflect the suggestion that LSTMs have the capacity to model the trace auxiliary input, but the increased performance of the LSTMs + trace agent would suggest that the two are modelling different things.
>
> We hope we were able to clear up some ambiguities in our work with both the clarifications and the changes to the paper. We think with all of these changes, the manuscript is much clearer and more concise.

---

### Review · Reviewer_usya · 2023-01-19

**Summary Of Contributions:**

This paper contributes a framework and empirical analysis for RL methods in partially-observed environments that rely on inputs outside of the current observation and action. The paper shows how a few methods in the literature fit into this framework, and analyzes efficacy in a few environments. The results indicate that using exponential traces as input to LSTMs are the most performant in the experiments in the larger environments.

**Audience:**

Yes

**Claims And Evidence:**

Yes

**Requested Changes:**

- Section 3.2.1 has a typo / sentence fragment that needs fixed: "For the Lobster environment, the preprocessing function g corresponds to a 1 if This particular form of history ...". Presumably, the "if" and the "This" should belong to difference sentences, with the "if"'s continuation currently missing.
- Eq. 10 seems incorrect. The $\tau$ summation variable is not used in the equation, and it sums from $t$ to $t$ (i.e. sums once). Please clarify or fix.
- Please change the labels of "Auxiliary Inputs" in Fig, 3 to something that clarifies their source (e.g. "Particle-Filtering Aux. Inputs"). In the submission, it is not clear from the Figure alone the method used to create auxiliary inputs.
- Please justify the absence of particle filtering results in Section 5. These environments seem small enough such that it may be possible to run a particle filter (define the transition and emission distributions). Consider extending the results with a particle filtering input.
- Please justify the limitation of the results to a single RL algorithm. Consider extending the results by using other RL algorithms, especially by those with existing open-source implementations and reference results on existing benchmarks.

**Strengths And Weaknesses:**

## Strengths
- The writing quality is very high, which makes the paper easy to understand.
- It is a good idea to unify some existing methods as a particular component of reinforcement learners in POMDPS for performing memory featurization, as it may guide other efforts to develop better implementations of this component.

## Weaknesses
- The results are limited to a small set of environments that appear to have been devised for use in this paper. No experimental results are presented on common RL benchmark environments. This significantly limits the practical takeaways for researchers who seek to improve performance on common benchmarks, or other, more general settings.
- The results are limited to only using a single RL algorithm, so we cannot draw conclusions about the relative efficacies of different auxiliary input methods.

---

> ### Author Response · Authors · 2023-02-03
> **Re: Review of Paper599 by Reviewer usya**
>
> Thank you for the kind words and feedback. We have taken into account the first three requested changes, and will be fixing the issues you highlighted in the manuscript. As for the final two requested changes, please see our comments/suggested clarifications below:
>
> **“Please justify the absence of particle filtering results in Section 5. These environments seem small enough such that it may be possible to run a particle filter (define the transition and emission distributions). Consider extending the results with a particle filtering input.”**
>
> The main goal of the experiments on the Fishing domains was to test the scalability of the trace auxiliary inputs, as well as how it interacts with and improves performance of neural networks and LSTMs. Despite looking relatively simple, the Fishing domains are in fact high-dimensional as we are operating in pixel-space. Moreover, in these experiments we make no assumptions about the agent having access to the underlying dynamics of the environment. With this lack of dynamics knowledge a particle-filter-based agent cannot update its particles at every time step. Additionally, even if it had access to the underlying model of the world, particle filtering would suffer from the curse of dimensionality in these environments, as the number of particles needs to increase quite a bit depending on the number of dimensions in the problem.
>
> That being said, we do agree with reviewer ***usya*** that understanding the relative performance gains induced by different techniques is interesting, and that's one of the reasons we compared and contrasted them in the Lobster environment (Figure 2a). These results suggest that in that particular environment, all these techniques have the same or similar effect of de-aliasing state and expanding the input space. Notice we don't claim the comparison between such techniques to be a contribution, we just claim to be demonstrating that all these different techniques can lead to similar effects in de-aliasing states to help value learning and policy representation.
>
> **“Please justify the limitation of the results to a single RL algorithm. Consider extending the results by using other RL algorithms, especially by those with existing open-source implementations and reference results on existing benchmarks.”**
>
> Thank you for the question. In order for us to understand the need for additional RL algorithms a more, we would like to clarify a few things with the reviewers and pose a few questions we should ask. Namely, what additional base RL-methods might we include, and what could we learn from such experiments. What potential hypothesis could we make here? Here is a selection of representative algorithms we could test:
>
> 1. policy gradient methods such as PPO or SAC
> 2. Model-based approaches like Dreamer
> 3. Asynchronous learning systems like A3C
>
> For 1, we should not expect policy gradient methods to somehow be able to learn in partially observable domains with a standard feedforward architecture. Indeed, the need for memory is agnostic to the choice of value-based method vs policy gradient method.
>
> In 2, for model-based RL, the agent learns both a policy and a world model. It is well-known that a major challenge in model-based RL is inaccurate models causing rollouts to produce invalid states and bad updates (Talivitie, 2016). This problem is only amplified more in partially observable domains; improving the representation used by the model via auxiliary tasks would be very promising indeed, but a massive and tangential quest when progress in MBRL has been, up until very recently, plagued with performance issues that are well beyond the scope of this paper.
>
> Finally, asynchronous methods with distributed actors still require good state information. Much of the work in this area is in fully observable deterministic domains like Mujoco. There certainly could be improvements from auxiliary tasks there and it would be nice to show. However, we are adhearing to a simple principle here: demonstrate the phenomenon and benefits clearly and defendable in a setting where we can reduce confounding factors and understand where the performance improvements are coming from. Adding more algorithms would add significant complexity for unclear gain.
>
> If the concern is that the benefit demonstrated here would not hold for other base RL algorithms we do not understand why this would be the case, and invite the reviewer to expand on the point.

---

> > ### Comment · Reviewer_usya · 2023-03-13
> > **Thanks for your response**
> >
> > ## Re: particle filtering suggestion
> > Okay, that's fine to leave as-is. Here's what I think, though:
> >
> > > in these experiments we make no assumptions about the agent having access to the underlying dynamics of the environment
> >
> > Sure, so evaluating a particle filter here would constitute an approach that does not satisfy the target assumptions, but nonetheless would be useful to include because it is approximating an optimal belief filter explicitly.
> >
> > > particle filtering would suffer from the curse of dimensionality in these environments
> >
> > This is plausible, but not a convincing reason not to run the experiment and show empirically that it's true.
> >
> > ## Re: single RL algorithm
> > The crux of your response is:
> > > However, we are adhearing to a simple principle here: demonstrate the phenomenon and benefits clearly and defendable in a setting where we can reduce confounding factors and understand where the performance improvements are coming from. Adding more algorithms would add significant complexity for unclear gain.
> >
> > I disagree that the gain would be unclear. The main gain is that we would learn is the practical generalizability of the paper's claims to popular RL algorithms. This is of particular interest to researchers and practitioners who use popular RL algorithms in POMDPs. It would be much easier for the paper to have impact if it presented results with the proposed method using popular RL algorithms in POMDPs. I think this is a clear gain.
> >
> > I also disagree that adding more algorithms would necessarily add significant complexity. It would require setting up and running additional experiments, but it would not add significant complexity to the results. I think most readers would understand (and many might expect) running the approach with multiple base RL algorithms), so from that perspective, adding more base RL algorithms to the result could actually simplify the results.
> >
> > > If the concern is that the benefit demonstrated here would not hold for other base RL algorithms we do not understand why this would be the case, and invite the reviewer to expand on the point.
> >
> > If this is a claim that you wish the paper to make, the onus is on the paper to provide evidence. The paper is weaker without it. Nevertheless, I'm happy to hypothesize why it may not hold in practice, but whether or not they are plausible hypotheses, my previous point holds. Here are a few I could think of (1) it may be too difficult to tune the hyperparameters of the other algorithms, such that the proposed method adds no value or (2) other RL algorithms may not suffer much in practice from the proposed problem phenomenon. The paper would be stronger if it showed that they do indeed suffer from it, and do indeed benefit from the proposed solution.

---

> > > ### Author Response · Authors · 2023-03-13
> > > **Setting up Lobster experiments**
> > >
> > > Thanks for taking the time to expand on your reasoning. We will set up additional experiments on the Lobster environment with a PPO-based agent.

---

> > > > ### Author Response · Authors · 2023-03-17
> > > > **Additional Lobster environment results with PPO**
> > > >
> > > > Thank you for the suggestion for a different base algorithm - it allowed us to also investigate and discuss the role of stochastic policies in the face of partial observability; we believe the paper is now better because of that. We've added results on the Lobster environment with PPO as the base algorithm as mentioned previously. With the suggested experiments, we've shown that auxiliary inputs are also effective when used by PPO, which learns stochastic policies.  The additional results (mentioned towards the end of Section 3.3, and elaborated on in Appendix C.5. Both changes are in blue text) not only allows us to quantify the effect of auxiliary tasks with an algorithm that is different from the value-based algorithm we used, but to also start to characterize the benefits of stochastic policies and auxiliary inputs on partially observable environments.

---

### Decision · Action_Editors · 2023-04-16

**Recommendation:** Accept as is

**Comment:**

The paper presents a united framework for how to construct auxiliary inputs for partially observed RL, connects this to the state construction literature, and shows useful empirical findings that could guide practitioners' choices.

**Audience:**

RL practitioners will use the investigation presented in this manuscript as a starting point to guide design decisions in handling partially observed settings, of which there are many of interest to the community.

**Claims And Evidence:**

This paper studies of a basic design choice in RL algorithms for partially observed tasks: in what format should "auxiliary" information about the past observations be fed as input to the RL agent? It studies three choices for this auxiliary input: exponentially decaying history traces, belief state estimates (based on a particle filtering method), and an approach that incorporates future predictions into the state.

Through drawing connections to work on state construction in RL, and empirical findings, the paper provides light on how to make this design choice. It verifies the state de-aliasing effect of auxiliary inputs, and also suggests that the simplest technique, exponential trace, works best.

While these findings are useful, there are a couple of limitations that limit the scope of these findings for future practitioners:
- Experiments here are in handful of relatively small environments designed to permit systematic study of the partial observability problem in RL.
- The experiments largely evaluate auxiliary inputs for a SARSA value-based RL algorithm, rather than for more widely used algorithms. Where this issue is fixed in appendices through experiments on PPO, the results are not comprehensive, the gaps between approaches are smaller, and the gains over having *no* auxiliary inputs are also smaller.